



# Nonlinear Time Series Analysis of Coastal Temperatures and El Niño–Southern Oscillation Events in the Eastern South Pacific

Berenice Rojo–Garibaldi[1], Manuel Contreras–López[2], Simone Giannerini[3], David Alberto Salas–de–León[4], Verónica Vázquez–Guerra[5], and Julyan H. E. Cartwright[6,7]

[1]Instituto de Ciencias Físicas, Universidad Nacional Autónoma de México, Av. Universidad s/n, Col. Chamilpa, Cuernavaca, Morelos, 62210
[2]Facultad de Ingeniería y Centro de Estudios Avanzados, Universidad de Playa Ancha, Playa Ancha 850, Valparaíso, Chile
[3]Dipartimento di Scienze Statistiche "Paolo Fortunati", Università di Bologna, Via delle Belle Arti 41, Bologna, 40126,Italy
[4]Instituto de Ciencias del Mar y Limnología, Universidad Nacional Autónoma de México, Av. Universidad 3000, Col. Copilco, México, 04510, México
[5]Posgrado de Ciencias del Mar y Limnología, Universidad Nacional Autónoma de México, Av. Universidad 3000, Col. Copilco, México, 04510, México
[6]Instituto Andaluz de Ciencias de la Tierra, CSIC—Universidad de Granada,Armilla, Granada, 18100, Spain
[7]Instituto Carlos I de Física Teórica y Computacional, Universidad de Granada, Armilla, 18071, Spain

**Correspondence:** Berenice Rojo-Garibaldi(berenice@icf.unam.mx)

**Abstract.** We investigate whether there are correlations between temperatures on the Eastern South Pacific coast, influenced by the Humboldt Current System, and El Niño–Southern Oscillation (ENSO) events, using a set of 16 oceanic and atmospheric temperature time series from Chilean coastal stations distributed between 18° and 45° S. Spectral analysis indicates periodicities that can be related to both internal and external forcing, involving not only ENSO, but also the Pacific Decadal Oscillation,

the Southern Annual Mode, the Quasi–Biennial Oscillation and the lunar nodal cycle. We carry out a nonlinear time series analysis. An asymptotic neural network test for chaos based on the largest global Lyapunov exponent indicates that the temperature dynamics along the Chilean coast is not chaotic. We calculate local Lyapunov exponents that characterize the short–term stability of the series. Using a cross entropy test, we find that just two stations in northern Chile, one oceanic, Iquique, and one atmospheric, Arica, present a significant positive correlation of local Lyapunov exponents with ENSO, with Iquique being the

station that displays very particular regional characteristics. This work, having a large–scale study area and using time series from hitherto unused sources (naval records), reveals the nonlinear dynamics of climate variability in Chile.

## 1   Introduction

The Peru–Chile or Humboldt Current System (HCS) is a notable phenomenon of the Eastern South Pacific coast. It is one of

the most biologically productive eastern border oceanic currents in the world (Pauly and Christensen, 1995) as a consequence of wind–driven coastal upwellings occurring at different intensity and frequency in the south-east Pacific (Daneri et al., 2000).



Such oceanic upwelling processes also impact atmospheric temperatures (Sobarzo et al., 2007), through either local or remote effects (Hormazabal et al., 2001). Atmospheric temperatures are further affected in this region by the warm (El Niño) and cold (La Niña) phases of El Niño–Southern Oscillation (ENSO) events (Vargas et al., 2007; Enfield and Allen, 1980; Pizarro and

Montecinos, 2004), and there is great interest in the possible influence of anthropogenic climate change (Sydeman et al., 2014).

A notable attempt to characterize the spatial and temporal patterns of atmospheric temperatures along the Pacific coast of South America was by Falvey and Garreaud (2009). They used in situ temperature records and satellite data for the period 1979–2006, and found that in northern and central Chile (latitudes 17°–37° S) atmospheric temperatures were cooling by about 0.20 °C per decade. They ascribed this cooling to a long–term La Niña phenomenon, which is consistent with the negative

trend observed in the Pacific Decadal Oscillation (PDO) for the same period. However, their analyses, and thus their results, are limited to the period after satellite data became available in the mid–1970s; that is, just when two major El Niño events, of 1982–83 and 1997–98, took place. The effects of these large disturbances on temperature trends based on linear analyses are not clear and must be considered to provide reliable, unbiased conclusions. This is especially true for studies conducted in highly dynamic regions like the Eastern South Pacific, where complex physical land–sea–atmosphere interactions suggest

nonlinear relationships between coastal upwelling, interdecadal variability of El Niño–type events and global warming (Vargas et al., 2007).

In a context of global warming, it is important to understand the spatial and temporal patterns of atmospheric temperatures in complex systems like the Eastern South Pacific region. Gaining knowledge on temperature changes, and in the phenomena behind them, is essential for understanding both scientific and societal issues. The nonlinear approach allows us to overcome

many of the limitations of the linear framework. Its foundations were laid in the early 1980s, when deterministic chaos became a very active field of research (Bradley and Kantz, 2015), including in the geosciences (see e.g. Zeng et al., 2015; Nicolis and Nicolis, 1984; Grassberger, 1986; Pierrehumbert, 1990; Vassiliadis et al., 1991; Lorenz, 1991; Cuculeanu and Lupu, 2001; Petkov et al., 2015). Elsner and Tsonis (1992) and Tsonis et al. (1993) proposed that the atmosphere can be seen as a weakly coupled system and that the finite dimension of the attractor found by some of these aforementioned studies could correspond

to the size of a subsystem. The dynamics of the atmosphere and the climate system are characterized by the property of sensitivity to initial conditions (Kalnay, 2003). This characteristic implies that a small error in observing the initial conditions will be exponentially amplified and this implies the impossibility of mid– and long–term forecasting. This property was already recognized in the first developments of weather forecasting (Thompson, 1963) and was associated with the nonlinear nature of deterministic dynamical systems by Lorenz (1963).

In this work we study the dynamics that governs the climatic variability of the South Eastern Pacific coast, its stabilities and instabilities. We discuss its consequences and its relationship with the HCS, the Pacific anticyclone, upwelling zones and the climatic phenomena that impact this region, thereby achieving a better understanding of the system's dynamics, impossible to achieve using only classical linear methods. To accomplish this task, we performed a single–panel nonlinear analysis of daily long series of ambient coastal and sea surface temperatures, distributed between 18° and 45° S and spaced every 3 to 4

degrees of latitude. We checked rigorously whether the systems that make up the study area exhibit chaotic dynamics, or not, through the largest Lyapunov exponent. In addition, we characterized the local stability of the systems using Local Lyapunov



Exponents (LLE), which quantify the growth of small perturbations generated by the internal variability. We examined how the dynamics are affected by ENSO events with a comparison of LLE and the Oceanic Niño Index in the different latitudes of the Chilean coast. Our study combines the periodicities of the time series, the type of dynamics involved and the bioclimatic characteristics found by latitude, in order to understand the role played by both internal and external forcing in the climatic variability of the Eastern South Pacific.

## 2 Data and Methods

### 2.1 Data pre-processing

Sea surface temperature (SST) data were obtained from the Hydrographic and Oceanographic Service of the Chilean Navy (SHOA), which has kept records for the main ports of the country since the mid 20th century. Data of atmospheric surface temperature (AST) beginning in the first half of the 20th century were provided by the Meteorological Directorate of Chile (DMC). Since the DMC weather stations are associated with the presence of airports in major cities, they usually coincide with the main ports. For all stations, the daily data were converted to weekly average temperature time series. In case of missing data, we applied Kalman filtering for imputation using the R package `imputeTS` (Moritz and Bartz-Beielstein, 2017). Tables 1 and 2 list the oceanic and atmospheric stations used in this study. The first column contains the location names that range from the northern to the southern regions of Chile and the second column indicates the time range; the third and fourth columns report the sample sizes.

### 2.1.1 Study area

Along the South Eastern Pacific littoral between 18° and 45° S there is a marked climatic gradient from an arid zone to a rainy Mediterranean zone. Three major divisions are identifiable along the coast: the north, 18°–30° S, center, 30°–37° S and south, 37°–45° S, of Chile (Sarricolea et al., 2017). Figure 1 shows the study area and the time series of both SST and AST used in this work.

Northern Chile is characterized by a narrow continental shelf, approximately 4–5 km wide, and an arid to semi-arid climate. Coastal upwelling is permanent in this region. Antofagasta stands out, as its orientation allows the presence of a pool of persistent hot water: a full-time El Niño (Piñones et al., 2007). La Serena, 30° S, presents a medium-sized continental shelf, approximately 20–30 km wide, where the surroundings show characteristics of persistent upwelling.

Central Chile has a wider continental shelf that extends about 40 km from the coast. It is characterized by a Mediterranean climate with four well-marked seasons. This system is under the influence of a coastal upwelling driven by the wind with a strong seasonal pattern and is one of the most productive regions of the ocean (Montero et al., 2007), with more than 50 % of the fish catch of Chile and 4 % of the world catch. Its hydrographical structure is strongly affected by the runoff of several rivers, whose discharges vary seasonally with a maximum during winter off Concepción, 36°49' S.



**Table 1.** Daily and weekly data for oceanic stations.

| Station | Time Period | Daily Data | Weekly Data |
|---|---|---|---|
| Arica [1] | 1951–1970 |  | 1015 |
|  | 1982–1999 | 25566 | 904 |
|  | 2004–2020 |  | 845 |
| Iquique | 1984–2020 | 13376 | 1910 |
| Antofagasta | 1946–2020 | 27392 | 3913 |
| Coquimbo | 1982–2020 | 14154 | 2022 |
| Valparaíso | 1961–2020 | 21914 | 3130 |
| Talcahuano [1] | 1949–1974 |  | 1356 |
|  | 1976–1989 | 26297 | 730 |
|  | 1991–2020 |  | 1543 |
| Corral | 1985–2020 | 13027 | 1861 |
| Puerto Montt | 1982–2020 | 14154 | 2022 |

[1] The data are split into three series, due to the presence of gaps.

**Table 2.** Daily and weekly data for atmospheric stations

| Station | Time Period | Daily Data | Weekly Data |
|---|---|---|---|
| Arica | 1960–2020 | 22281 | 3183 |
| Iquique | 1981–2020 | 14520 | 2074 |
| Antofagasta | 1960–2020 | 22281 | 3183 |
| Serena | 1973–2020 | 17461 | 2494 |
| Rodelillo [2] | 1971–1995 | 18232 | 1300 |
|  | 2002–2020 |  | 991 |
| Concepción | 1970–2020 | 18597 | 2656 |
| Valdivia | 1968–2020 | 19268 | 2752 |
| Puerto Montt | 1970–2020 | 18597 | 2656 |

[2] The data are split into two series, due to the presence of gaps.





Southern Chile includes the northern limit of Chilean Patagonia, one the most extensive fjord regions in the world. The climate is rainy. It covers almost 240 000 km² with an extremely complex geomorphology in one of the least densely populated areas of the country. Water circulation essentially follows a two–layered estuarine flow pattern. One superficial layer of brackish

85    water is fed by rivers and glaciers and another marine layer, mainly Subantarctic Surface Water (SAAW), enters predominantly through a subsurface layer (Sievers and Silva, 2008).

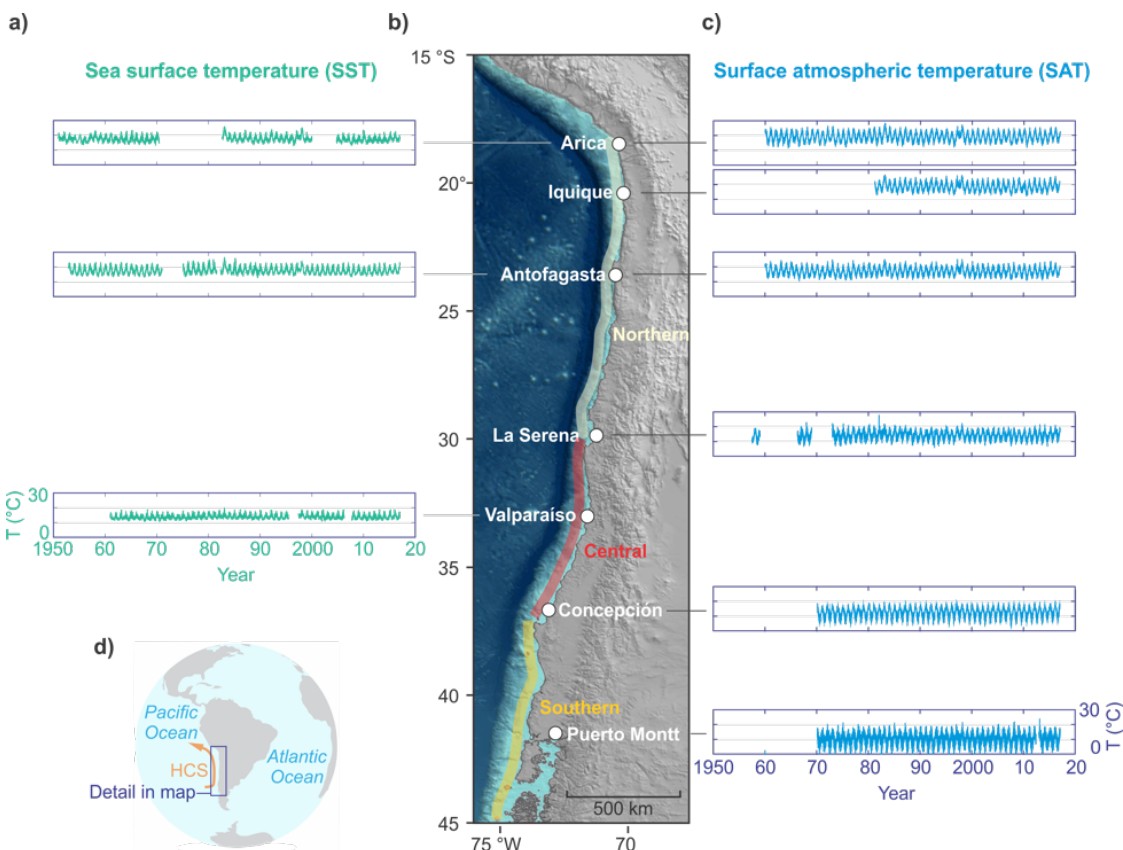

**Figure 1.** Study area and time series used in this work. a) Sea surface temperature for stations from northern to southern Chile. b) Study area. c) Atmospheric surface temperature for the same region. d) Study area relative to the Humboldt Current System (HCS). For comparative purposes, all the time series plots here use the same temperature and time scales.





### 2.1.2 Trend estimation

We assume that the series $X_t$, $t = 1, \ldots, n$ admits the following decomposition:

$$X_t = m(t/n) + z_t, \tag{1}$$

$$z_t = \sigma_t u_t,$$

$$u_t = \sum_{j=0}^{\infty} \psi_j \varepsilon_{t-j}, \quad \psi_0 = 1,$$

where $\{\varepsilon_t\}$ is a zero-mean i.i.d. sequence with constant variance and finite fourth moments, $m(\cdot)$ is a smooth deterministic trend function, $z_t = \sigma_t u_t$ is a stochastic process with $\sigma_t$ accounting for unconditional heteroskedasticity and $u_t$ is a stationary linear process. We estimate $m(\cdot)$ by means of a local polynomial smoother (loess). For the estimation of the long-run trend the degree of the smoother has been set to one to mitigate boundary effects, otherwise the order is two. The confidence bands at level 95% for the nonparametric estimator of the trend have been derived using an autoregressive wild bootstrap scheme:

1. obtain the estimate of the trend $\hat{m}(t)$ using the bandwidth $\tilde{h}$ and derive the residuals $\hat{z}_t = X_t - \hat{m}(t)$;

2. derive the bootstrap errors $z_t^* = \xi_t^* \hat{z}_t$, where $\xi_t^* = \gamma \xi_{t-1}^* + \nu_t^*$, $\nu_t^* \sim$ i.i.d. $N(0, 1 - \gamma^2)$ and $\gamma = \theta^{1/l}$, with $\theta = 0.1$ and $l = 1.75 \sqrt[3]{n}$.

3. build the bootstrap series $X_t^* = \hat{m}(t/n) + z_t^*$ for $t = 1, \ldots, n$ and estimate the trend $\hat{m}^*(t)$ upon it using the same bandwidth $\tilde{h}$ as in step 1.

The scheme provides valid confidence intervals under the assumptions as in Eq. (1) where the detrended series is dependent. For more details see Friedrich et al. (2020).

### 2.2 Chaos and the largest Lyapunov exponent

The largest global Lyapunov exponent quantifies the sensitivity to initial conditions and is one of the hallmarks of the presence of chaos. Assume that the series is generated by the following dynamical system in $\mathbb{R}^m$:

$$X_t = F(X_{t-1}), \qquad X_t \in \mathbb{R}^m, \tag{2}$$

and let $X_0$ and $X_0'$ be two close initial conditions and $X_n$ and $X_n'$ their value after $n$ time steps, respectively. Then

$$\left\| X_n - X_n' \right\| \approx \exp(n\,\lambda_1) \left\| X_0 - X_0' \right\|$$

where $\|\cdot\|$ is a suitable norm and $\delta = \left\| X_0 - X_0' \right\|$ is a small perturbation. Hence

$$\lambda_1 = \lim_{n \to \infty} \lim_{\delta \to 0} \frac{1}{n} \ln \left( \frac{\left\| X_n - X_n' \right\|}{\left\| X_0 - X_0' \right\|} \right) \tag{3}$$





is the global largest Lyapunov exponent. Note that

$$X_n - X_n^{'} \approx \delta \cdot DF^{(n)}(X_0) = \delta \prod_{t=0}^{M-1} DF(X_t) = \delta \prod_{t=0}^{M-1} J_{M-t},$$

where $F^{(n)}(X_0)$ is the $n$-fold iteration of the map $F$ and $J_t$ is the Jacobian of the map $F$ evaluated at $X_t$. Hence

$$\lambda_1 = \lim_{M \to \infty} \frac{1}{2M} \ln \nu_1 \left( \mathbf{T}'_M \mathbf{T}_M \right), \tag{4}$$

where $\mathbf{T}_M = \prod_{t=0}^{M-1} J_{M-t}$ and $\nu_1(A)$ is the largest eigenvalue of the matrix $A$. $\lambda_1$ measures the average rate of divergence
115  of nearby starting trajectories and chaotic dynamics implies $\lambda_1 > 0$. The two equivalent definitions of the largest Lyapunov
exponent given in Eq. (3) and Eq. (4) are reflected in the two approaches to estimating it in finite samples. The first approach
refers to Eq. (3) and is called *direct* since it finds close pairs of state vectors and measures the average divergence of trajectories
over time (Rosenstein et al., 1993; Kantz, 1994). Typically the logarithm of such a divergence is plotted over time and, if it
is possible to identify a linear scaling region, its slope is the direct estimator of $\lambda_1$. Typically, the exercise is repeated for a
120  range of embedding dimensions and time delays to assess the reliability of the estimate. One problem with this approach is that
it is sensitive to noise: it cannot distinguish between divergence due to noise from exponential divergence due to the chaotic
dynamics (Giannerini and Rosa, 2004). Moreover, there are no results on the asymptotic distribution of the estimator so that it
is not possible to build a proper statistical test for the hypothesis of chaos. In Giannerini and Rosa (2001) a bootstrap solution
for continuous time processes was proposed, but the intrinsic sensitivity to noise of the direct estimator remains, so that we
125  only use it to cross-check the result from the Jacobian based estimator which is based upon a neural network model of the map
$F$ and its Jacobian $J$; refer to Eq. (4). The modelling step allows us to filter out the effect of noise, obtain consistent estimators
and perform a proper statistical test for chaos (Shintani and Linton, 2004). The null hypothesis tests $H_0 : \lambda_1 \leq 0$ against the
alternative hypothesis of a positive exponent $H_1 : \lambda_1 > 0$. The asymptotic, one sided, test is based upon the standardized
statistic

130  $$Z = \frac{\hat{\lambda}_M}{\sqrt{\hat{\mathbf{\Phi}}/M}} \xrightarrow[M \to \infty]{d} N(0,1) \tag{5}$$

where $M$ is the number of steps ahead and $\hat{\mathbf{\Phi}}$ is a HAC estimator of the asymptotic variance of the $M$-step local Lyapunov
exponent estimator $\hat{\lambda}_m$. The null hypothesis is rejected at level $\alpha$ if the value of the statistic $Z^*$ exceeds $z_\alpha$, the critical value
of the standard Gaussian random variable.

## 2.3  Local Lyapunov exponents

135  A positive largest global Lyapunov exponent indicates sensitive dependence on initial conditions and is an important theoretical,
global indicator of the presence of chaos. However, it is defined in the double limit of infinitesimal perturbations and infinite
time steps ahead; hence it poses no practical limit to the predictability of a dynamical system (Ziehmann et al., 2000; Giannerini



and Rosa, 2004), which may typically have state space regions of enhanced predictability, coupled with zones of instability. Nonlinear dynamical systems, be they chaotic or not, are characterised by this non–uniform predictability and one way of quantifying it is through local (finite–time) Lyapunov exponents (LLEs). In this respect, local Lyapunov exponents can be used as a time series tool that provides different, complementary, information with respect to global Lyapunov exponents (Bailey et al., 1997).

The analysis involves examining the short term predictability, or unpredictability, of a system and identifying the regions in state space where they occur (Abarbanel et al., 1992; Smith, 1992; Wolff, 1992). Suppose that the data $\{x_t\}$ are from a time series generated by a nonlinear autoregressive model

$$x_{t+1} = f(x_t, x_{t-1}, \ldots, x_{t-d+1}) + \varepsilon_t, \tag{6}$$

where $x_t \in \mathbb{R}$ and $\{\varepsilon_t\}$ is a sequence of independent random variables or perturbations with $E(\varepsilon_t) = 0$ and $\text{Var}(\varepsilon_t) = \sigma^2$. It is important to note that the error in Eq. (6) is not measurement error, but dynamical noise, an inherent part of the dynamics of the system. The state–space representation of the system is

$$X_{t+1} = F(X_t) + e_t, \tag{7}$$

where $X_t = (x_t, x_{t-1}, \ldots, x_{t-d+1})$ and $e_t = (\varepsilon_t, 0, 0, \ldots, 0)$ are in $\mathbb{R}^d$.

The LLE is defined by making a small perturbation of $X_t$ to $X_t^*$ and following forward the perturbed and unperturbed trajectories. The LLE measures the difference between the two trajectories after $m$ time steps

$$\lambda_M(t) = \frac{1}{M} \log \frac{\left\| X_{t+M} - X_{t+M}^* \right\|}{\left\| X_t - X_t^* \right\|}. \tag{8}$$

If we choose to perturb only the first component of $X_t$ and follow the growth of that perturbation along the actual sample path, this corresponds to a perturbation of Eq. (6) at time $t$, i.e., perturbing $x_t$ and the following the growth of that single perturbation along the actual sample path. This definition of the LLE is related to noise amplification by the system, and thus to predictability (Yao and Tong, 1994). Let $\sigma_M^2(x) = \text{Var}(X_{t+M} \mid X_t = x)$. In a scalar system with small additive noise,

$$\sigma_M^2(x) = \sigma^2 \mu_M(x)(1 + o(1)),$$

where $\sigma^2 = \text{Var}(\varepsilon_t)$ and

$$\mu_M(x) = 1 + \sum_{j=1}^{M-1} \left\{ \prod_{k=j}^{M-1} \dot{f}(f^{(k)}(x)) \right\}^2. \tag{9}$$

This can be re-written as

$$\mu_M(x) = 1 + \sum_{j=1}^{M-1} \exp\left\{ 2(M-j)\lambda_{M-j}(f^{(j)}(x)) \right\} \tag{10}$$





where $\lambda_j$ are LLEs along the "skeleton" system with the noise deleted (Yao and Tong, 1994). However, to leading order in $\sigma$ in this expansion, the skeleton LLEs agree with those along the sample path for any fixed finite $M$. For a system with dynamical noise the values of $\mu_M$ set the limits to large-sample forecasting accuracy (Tong, 1995). Thus Eq. (10) shows a close link — for small noise — between LLEs and local unpredictability.

As perturbations become small, $\lambda_M(t)$ will depend on derivatives of the map $F$. Then

$$\lambda_M(t) = \frac{1}{M} \ln \|J_{M+t-1}J_{M+t-2}, \dots, J_t U_0\|, \tag{11}$$

where $J_t$ (sometimes written $J(X_t)$) is the Jacobian matrix of F evaluated at $X_t$, $U_0$ is a unit vector and $\|\cdot\|$ is the Euclidean vector norm. Since $\lambda_m(t)$ is a function of time, the LLE depends on the trajectory and can be thought of as an $M$-*step ahead* local Lyapunov exponent process. If $U_0$ is chosen at random with respect to the uniform measure on the unit sphere, then with probability 1 as $M \to \infty$, $\lambda_M(t)$ converges to the global Lyapunov exponent, because $U_0$ has zero probability of falling into a subspace corresponding to subdominant exponents. In practice, we usually take $U_0 = (1, 0, 0, \dots, 0)$ and the estimators for local exponents are the same as for global ones, with the notable difference that the latter depend upon the state space location.

## 3 Results

We carried out a spectral analysis for the 16 stations, 8 atmospheric and 8 oceanic. In Figure 2, we show the results for three atmospheric and three oceanic stations, which represent the northern, central and southern regions of Chile. The complete set of results is in Appendix A2.

The oceanic stations that show spectral peaks with the longest periodicities are Valparaíso with 27 years; Talcahuano with 20 years; Antofagasta with 18 years; and Arica with 14 years. Four stations have periodicities with an intermediate value, 6 to 10.6 years. Three stations have periodicity around 4 years (Antofagasta, Valparaíso and Talcahuano). Two stations have peaks around 3 years, and six stations present 2 year periods. In the eight stations periodicities between 1.3 and 2 years were also found. Most stations show periods of around 1.5 years.

The atmospheric stations that show spectral peaks with the longest periodicities are Antofagasta and Serena with 20 years; Arica with 19 years; Iquique with 18.8 years; Concepción with 16.4 years and Puerto Montt with 16 years. Furthermore, Arica and Antofagasta share a periodicity of 5.1 years. There are 3 stations with periodicity around 4 years, and 5 stations with periodicity around 3 years. Periodicities around 1.5 years were also found, as in the oceanic case. In general, longer periodicities can be observed for atmospheric surface temperature time series than for sea surface temperature. The possible explanation of these periodicities can be found in the discussion in Section 4.1.

We estimated the trend by means of a local polynomial smoother (loess) as detailed in Section 2.1.2. For the estimation of the long-term trend the degree of the polynomial was set to one to mitigate boundary effects, otherwise the order is two. The confidence bands at level 95 % for the non-parametric estimator of the trend were derived using an autoregressive wild bootstrap scheme, which is robust against the presence of dependence and heteroskedasticity. We show the results for Antofagasta (SST, AST) and Iquique (SST, AST) in Fig. 3. The estimated long-term trend for Antofagasta and Iquique might seem to indicate



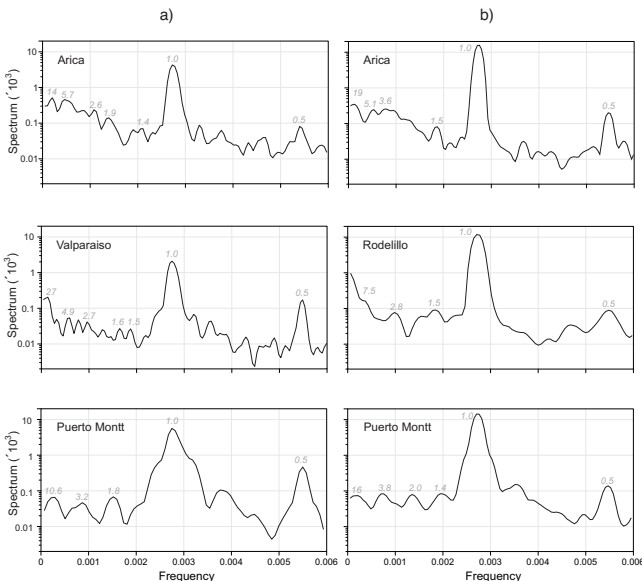

**Figure 2.** Spectral analysis for three representative stations from the northern, central and southern regions of Chile. Column a) corresponds to sea surface temperature time series (oceanic stations) and column b) to atmospheric surface temperature time series (atmospheric stations). Dominant peaks are indicated together with their associated periods in years.

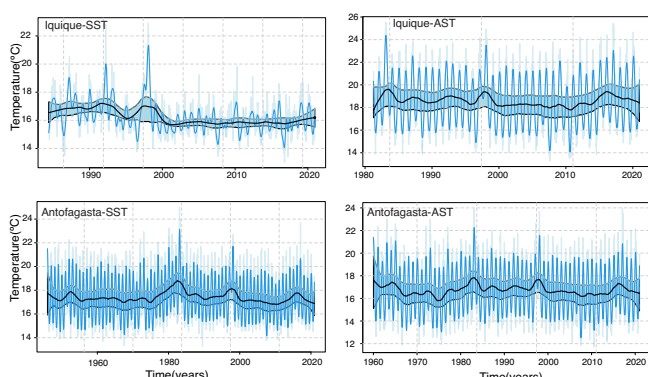

**Figure 3.** Temperature time series of Antofagasta and Iquique. Top left: Antofagasta sea surface temperature; top right: Antofagasta atmospheric surface temperature. Bottom left: Iquique sea surface temperature; bottom right: Iquique atmospheric surface temperature. Interpolated original (light blue), trend–cycle (blue) and long-term trend (black), together with confidence bands at 95 % (light blue).

some sort of tendency, but this is ruled out if we take into account the uncertainty (confidence bands in light blue). This suggests that the process that has generated the time series for both stations is mean-stationary and this seems to hold for all the stations. For more details, see Appendix A3.





**Table 3.** Output from the best neural network fit from the grid search test for chaos of the time series of the oceanic stations. The table reports the embedding dimension $d$, the time delay $\tau$, the number of hidden units $k$, the value of the minimised BIC, the global Lyapunov exponent $\hat{\lambda}_1$ together with the value of the test statistic $Z^*$ and the $p$-value of the test.

| Station | Time period | $d$ | $\tau$ | $k$ | BIC | $\hat{\lambda}_1$ | $Z^*$ | $p$-value |
|---|---|---|---|---|---|---|---|---|
| Arica | 1951–1970 | 10 | 1 | 1 | -811.84 | -0.045 | -22.05 | 1 |
| Arica | 1982–1999 | 11 | 4 | 11 | -21.33 | -0.027 | -28.45 | 1 |
| Arica | 2004–2020 | 8 | 1 | 8 | -509.49 | -0.114 | -22.79 | 1 |
| Iquique | 1984–2020 | 3 | 1 | 3 | -2094.32 | -0.083 | -34.38 | 1 |
| Antofagasta | 1946–2020 | 12 | 1 | 1 | -4298.17 | -0.017 | -25.81 | 1 |
| Coquimbo | 1982–2020 | 11 | 1 | 11 | -1281.49 | -0.041 | -28.06 | 1 |
| Valparaíso | 1961–2020 | 12 | 1 | 12 | -2918.70 | -0.056 | -37.36 | 1 |
| Talcahuano | 1949–1974 | 12 | 1 | 3 | -853.68 | -0.040 | -12.79 | 1 |
| Talcahuano | 1976–1989 | 12 | 1 | 1 | -215.38 | -0.050 | -33.62 | 1 |
| Talcahuano | 1991–2020 | 12 | 4 | 12 | 165.70 | -0.054 | -12.36 | 1 |
| Corral | 1985–2020 | 12 | 1 | 12 | -668.83 | -0.035 | -15.95 | 1 |
| Puerto Montt | 1982–2020 | 11 | 1 | 59 | -364.38 | -0.032 | -19.92 | 1 |

We estimated the largest Lyapunov exponent of the series by means of a single–layer, feed forward, neural network model of
the map and used it to test the null hypothesis of no chaos ($H_0 : \lambda_1 \leq 0$). We chose the best model by means of the generalized
Bayesian Information Criterion (BIC), where the global minimum is searched on a grid of embedding dimensions from $d = 3$
to $d = 20$ and time delays from $\tau = 1$ to $\tau = 4$. The results are presented in Tables 3 and 4 for the oceanic and atmospheric
stations respectively. Besides the station name and the time period, the tables report the values of the embedding dimension $d$,
time delay $\tau$ and number of hidden units $k$ of the network that produce the minimised BIC criterion over the parameter grid.
The last two columns contain the value of the test statistic $Z^*$ and the $p$-value of the test based on the asymptotic Gaussian null
distribution, respectively (Shintani and Linton, 2004). All the estimated exponents are negative and the test never rejects the
null hypothesis of absence of chaos, both for the oceanic and the atmospheric stations.

This result is an important indication that the dynamics behind temperature in Chile is most probably nonlinear but not
chaotic (Lorenz, 1963). We use local Lyapunov exponents (LLEs) to characterise local predictability throughout the stations.
We varied the number of steps ahead $M$ from 4 to 52 (from 1 month to 1 year) for oceanic stations and up to 260 steps
ahead (5 years) for atmospheric stations. In order to uncover any association with El Niño, we chose three specific windows
corresponding to the short (0 to 3 months), medium (6 to 9 months) and long term (12 to 22 months) and paid special attention
to the results obtained for the 1, 6 and 12 month windows. In Figures A3 to A5 of the Appendix we show the time series
of the oceanic stations where the information on the 1 month, 6 months and 1 year LLEs, respectively, is superimposed: red
corresponds to a positive LLE whereas green indicates a negative LLE. The same information for the atmospheric stations is





**Table 4.** Output from the best neural network fit from the grid search test for chaos of the time series of the atmospheric stations. The table reports the embedding dimension $d$, the time delay $\tau$, the number of hidden units $k$, the value of the minimized BIC, the global Lyapunov exponent $\hat{\lambda}_1$ together with the value of the test statistic $Z^*$ and the $p$-value of the test.

| Station | Time period | $d$ | $\tau$ | $k$ | BIC | $\hat{\lambda}_1$ | $Z^*$ | $p$-value |
|---|---|---|---|---|---|---|---|---|
| Arica | 1960–2020 | 12 | 1 | 12 | -3170.93 | -0.014 | -15.46 | 1 |
| Iquique | 1981–2020 | 12 | 1 | 12 | -1685.18 | -0.013 | -13.21 | 1 |
| Antofagasta | 1960–2020 | 13 | 1 | 12 | -2804.60 | -0.014 | -13.05 | 1 |
| Serena | 1973–2020 | 17 | 1 | 17 | -871.98 | -0.013 | -9.53 | 1 |
| Rodelillo | 1971–1995 | 19 | 4 | 79 | 1325.28 | -0.016 | -13.73 | 1 |
| Rodelillo | 1973–2020 | 18 | 3 | 58 | 1065.62 | -0.012 | -8.42 | 1 |
| Concepción | 1970–2020 | 20 | 1 | 20 | 1373.17 | -0.010 | -12.06 | 1 |
| Valdivia | 1968–2020 | 18 | 1 | 18 | 2950.07 | -0.007 | -7.81 | 1 |
| PuertoMontt | 1970–2020 | 19 | 4 | 79 | 2337.82 | -0.012 | -8.09 | 1 |

reported in Fig. A6 to A8. Likewise, in Fig. 4 and 5, we present the phase space portraits of the attractors for the stations with superimposed information on the LLEs in red and green.

When comparing the LLEs of the oceanic stations with their atmospheric counterparts for the different windows, we found that for the 1 month window, both classes of stations showed unstable behavior throughout the record, except for Iquique–

SST that shows both stability and instability patterns. For the 6 month window in SST and AST, mixed stability/instability is observed, except for Iquique, where there is stability throughout the record. In the Antofagasta and Puerto Montt SSTs similar behavior can be observed, showing instability at the extremes and stability in the middle part. Finally, for the 1 year window, all oceanic stations showed stability throughout the record, except for Talcahuano and Puerto Montt, that is, the southern region. Of the atmospheric stations, Arica, Iquique and Antofagasta showed instability in the extremes and stability in the middle part

of the record; Rodelillo behaves totally stable, while Serena, Concepción and Valdivia present totally stable or unstable parts throughout the series, this is not the case for Puerto Montt, which presents very little instability.

Boxplots of the LLEs versus steps ahead are shown in Fig. 6 and 7. Clearly, the sea surface temperature series approach the global exponent faster than the atmospheric series. For instance, for $M = 28$ steps ahead, the boxplots of the LLEs are already negative for all the SST series, with the exception of Puerto Montt; Fig. 6. On the contrary, this does not happen for any of

the AST series; Fig. 7. Overall, the results indicate a qualitative difference in the temperature dynamics of the two types of series. This can be also explained by considering that, on average, $\hat{\lambda}_1 \approx -0.05$ for the sea surface series but $\hat{\lambda}_1 \approx -0.01$ for the atmospheric series, so the latter are globally more stable. The presence of the ocean and its exchanges of heat and momentum with the atmosphere can easily reduce the instability properties of the atmospheric flow (Vannitsem et al., 2015).

We can compare the information conveyed by local Lyapunov exponents with the Oceanic Niño Index (ONI). El Niño is

characterized by a positive ONI greater than or equal to +0.5 °C. La Niña is characterized by a negative ONI, less than or equal





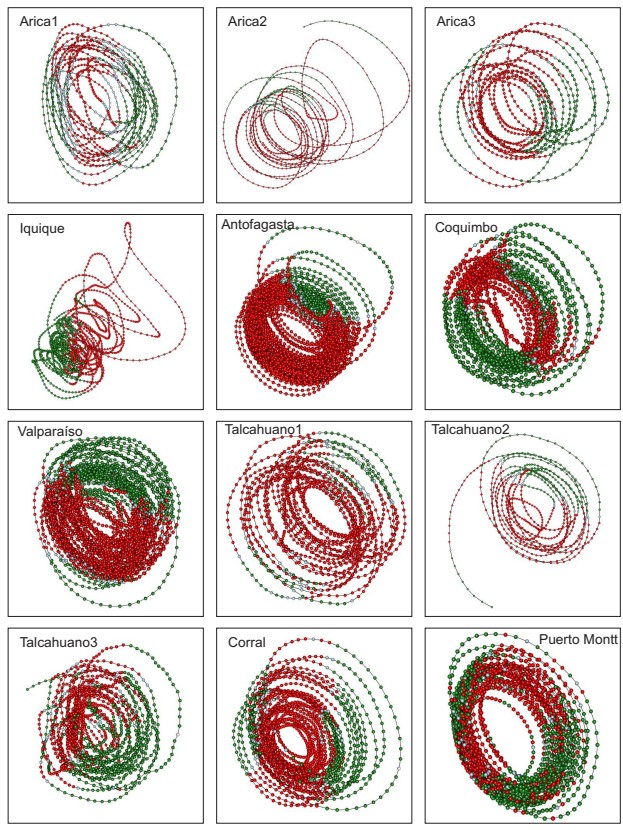

**Figure 4.** Attractors for sea surface temperature with superimposed information on the local Lyapunov exponents. Red corresponds to a positive LLE whereas green indicates a negative LLE. The subsections indicate the latitudinal order of the stations, in the case of Arica and Talcahuano there are three courts, which are numbered from 1 to 3.The temporary windows are: 1 month for Iquique; 4 months for Arica; 5 months for Coquimbo, Valparaiso and Talcahuano and 6 months for Antofagasta, Corral and Puerto Montt.

to -0.5 °C. To be classified as a full–fledged El Niño or La Niña episode, these thresholds must be exceeded for a period of at least 5 consecutive overlapping 3 month seasons. The Climate Prediction Center of the US National Oceanic and Atmospheric Administration (NOAA) considers El Niño or La Niña conditions to occur when the monthly Niño 3.4 SST departures meet or exceed +/-0.5 °C along with consistent atmospheric features. These anomalies must also be forecast and should persist for 3

consecutive months.

In the case of the oceanic stations, we may note the following by a visual comparison: in Antofagasta the expected behavior is observed, that is, the whole series shows instability for the 1 and 6 month windows, always showing the same behavior throughout the record. This may be due to the warm water pool that is always found here simulating an El Niño. The same pattern is observed for the 1 year window, only this time the behavior is completely stable. Arica shows instability throughout

the record for the 1 month window, where instability decreases markedly during El Niño events of 1953–54, 1958–59, 1963–





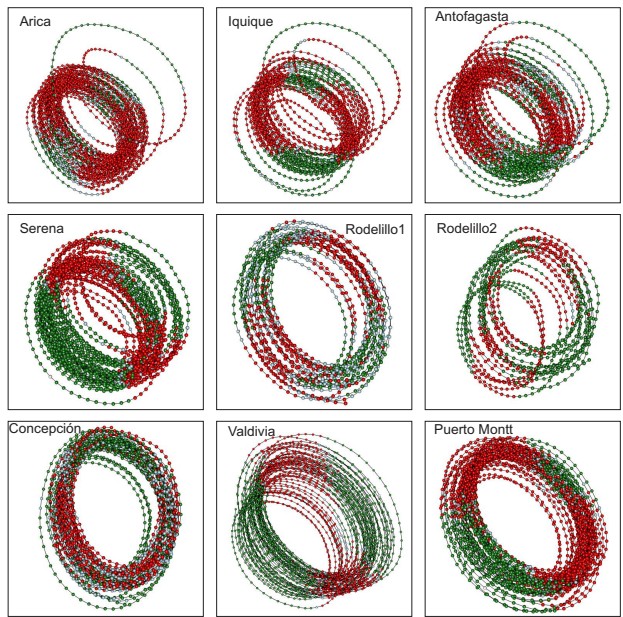

**Figure 5.** Attractors for atmospheric surface temperature with superimposed information on the local Lyapunov exponents. Red corresponds to a positive LLE whereas green indicates a negative LLE. The subsections indicate the latitudinal order of the stations, for the case of Rodelillo, these are two courts. The temporary windows are 5 months for Serena, Rodelillo, Concepción and Valdivia and 6 months for Arica, Iquique, Antofagasta and Puerto Montt.

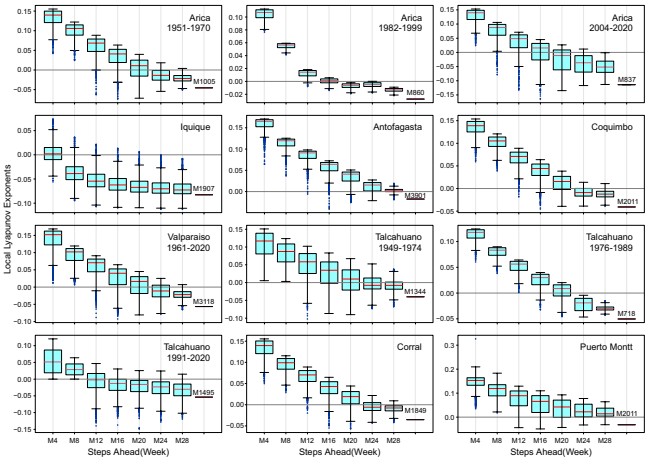

**Figure 6.** Boxplots of the local Lyapunov exponents $\hat{\lambda}_M$ versus steps ahead ($M$ in weeks) for the sea surface temperature series. The series are arranged latitudinally.



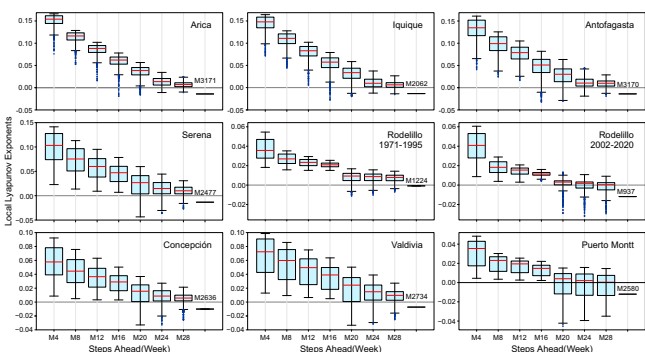

**Figure 7.** Boxplots of the local Lyapunov exponents $\hat{\lambda}_M$ versus steps ahead ($M$ in weeks) for the atmospheric surface temperature series. The series are arranged latitudinally.

64, 1965–66, 2009–10 and 2014–16. During the 6 month and 1 year windows, it becomes more stable than unstable and shares the same behavior with Talcahuano, where the third cut can be located lower than the other two and a very marked variability in the LLE can be seen. In these two stations the trajectory of the phase space for the second cut is interrupted. In Talcahuano, interesting behavior is only observed, with respect to the ONI, for the 1 month window, where the instability for the first cut,

1949–1974, decreases during El Niño events of 1953–54, 1958–59, 1963–64, 1965–66, 1968–69 and 1972–73, to show rising behavior in the third cut of 1991–2020 during El Niño events of 1991–92, 1994–95, 1997–98; 2002–03, 2004–05, 2009–10 and 2015–16. In the case of Iquique, the 1 month and 1 year windows have stable/unstable behavior, it is the only series in which the peaks of both the stable and unstable part coincide with several El Niño events of 1987–88, 1991–92, 1994–95, 1997–98, 2002–03, 2004–05, 2006–07, 2009–10, 2014–16, 2018–19, 2019–20. For the 6 month window, the behavior is stable

throughout the record, but the stability peaks continue to coincide with the same El Niño events, with the exception of 2002–03, 2004–05, 2006–07, 2018–19 and 2019–20. Finally, Valparaíso, in the 1 month window, shows a decrease in the unstable part that corresponds to El Niño events of 1968–69, 1972–73, 1982–83, 1997–98 and 2014–16.

In the case of the atmospheric stations, that in Antofagasta, the 1 month window shows unstable behavior throughout the entire record and shows a marked decrease during the 1972–73, 1982–83 and 1997–98 El Niño events. In the case of Arica,

for the same window, a decrease in the unstable part is observed for the same El Niño events plus that of 2014–16. For its part, Puerto Montt, in its 1 and 6 month windows, behaves in the same way as Antofagasta in its oceanic record, that is, it shows continuous behavior throughout the entire time series. The same occurs with Rodelillo for the 1 month window; its behavior is similar to that of Antofagasta and Puerto Montt. Finally, for the 1 year window in Serena, instability seems to increase from 1971 to 1982–83, which is where it coincides with the El Niño event of that year, the behavior decreases and increases again

to coincide with El Niño of 1987–88 and likewise, a peak of instability coincides with El Niño of 2014–16.

In summary, in the case of the oceanic stations, in Arica, Valparaíso and Talcahuano, the unstable part decreases significantly during various El Niño events, while for Antofagasta the behavior is continuous throughout the record, both for the stable and the unstable part; on the other hand, in Iquique the unstable behavior increases during El Niño events and the stability decreases



after them. For the atmospheric stations, it is in Arica, Iquique and Antofagasta where a decrease in instability during several
El Niño events can be seen, contrary to what Serena shows with an increase in instability, only that the latter sample for the 1
year window, while the previous stations show an interesting behavior of decrease for the 1 month window.

In order to go beyond this visual comparison and to corroborate whether there is a relationship between the ONI index
and the dynamics of climate variability in Chile seen through the LLEs, we computed the cross entropy measure proposed in
Giannerini et al. (2015) and Giannerini and Goracci (2023); see Appendix A5. Rejection bands were calculated with a moving
block bootstrap and series were pre–whitened — otherwise serial correlation will create false positive rejection — using an
autoregressive filter. The test was performed for all stations before and after pre–whitening. The results of this test for the
stations with a significant cross-correlation are shown in Fig. 8; for the results of the rest of the stations see Appendix A5.

The cross-entropy test indicates a significant dependence at lag $+1$ for Iquique–SST with a 1 month time horizon; in the
case of 6 months for the same station the lag was $-2$ and for the atmospheric stations only Arica–AST showed a significant
dependence on lags $+2$ and $-3$. This implies that the ONI index at time $t$ is correlated with the LLE at times $t+1$, $t-2$,
$t+2$ and $t-3$, respectively, for the aforementioned stations. For the 6 month time horizon, Corral also presented a significant
autocorrelation, which is likely a false positive; see Fig. A14. We also applied the test to correlate the ONI index and 1 year–
ahead LLEs for Serena. The results in Fig. A17 do not present any significant correlations. We discuss the positive results for
Iquique–SST and Arica–AST below.

## 4  Discussion

### 4.1  Spectral Analysis

The periods of 18, 18.8 and 19 years corresponding to Antofagasta–SST, Iquique–AST and Arica–AST, although they could
be related to the PDO, might also be associated with the lunar nodal cycle of 18.61 years, popularly known as "lunar wobble"
(this low–frequency tidal cycle is due to the oscillation of the maximum lunar declination with respect to the equator (Rossiter,
1962; Black et al., 2009)), since the variations of the lunar cycle influence the movement of ocean currents, which move hot or
cold water around the Earth.

The 4-year periods found in Antofagasta, Valparaíso and Talcahuano may be related to the lunar perigee subharmonic of
4.4 years. The external forcing caused by the Moon is very important not only because of the changes caused in interannual
time scales to the variability of the tidal range, but also because it influences the hydrodynamic energy of the ocean, altering
sedimentation patterns (Oost et al., 1993; Gratiot et al., 2008) and coastal erosion (Smith et al., 2007), water level and coastal
flood risk (Thompson et al., 2021), as well as ocean stratification (Devlin et al., 2017b, a, 2019).

Interannual periods can also be observed in the oceanic stations, 2.7 years for Iquique and Valparaíso; 2.6 for Arica; 2.5 for
Coquimbo and Talcahuano; 2.4 for Antofagasta; 2.0 for Corral and 1.8 for Puerto Montt. In the case of atmospheric stations,
2.7 years was found for Serena; 2.8 for Rodelillo; 2.3 for Valdivia and 2.0 for Puerto Montt. These periods could be related to
the Quasi–Biennial Oscillation (QBO), a climatic alteration located in the Pacific Ocean that consists of zonally symmetrical
easterly and westerly wind regimes that regularly alternate with a period that varies between 24 and 30 months and is manifested

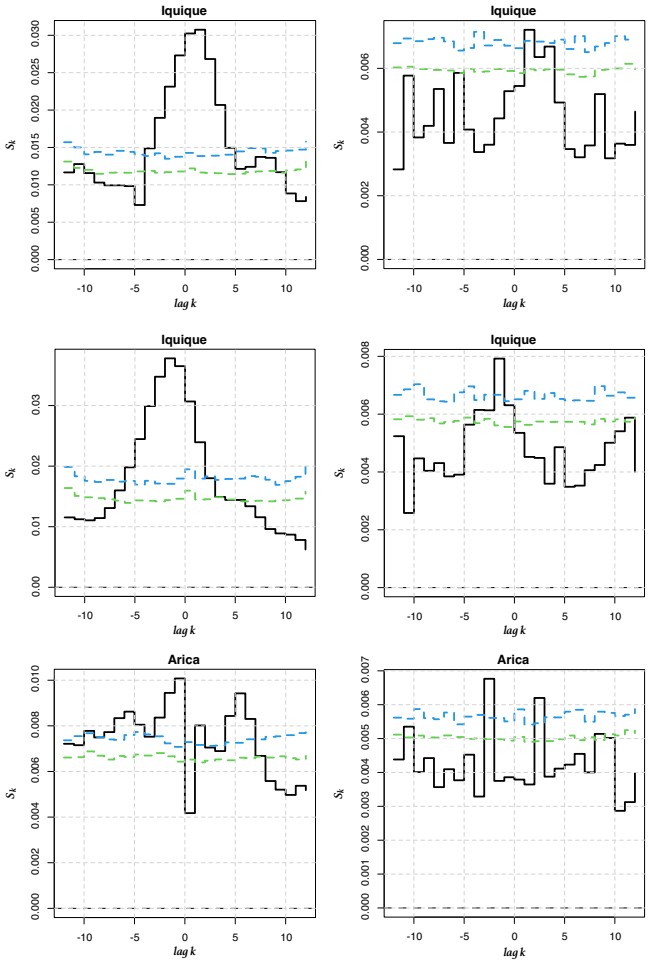

**Figure 8.** Cross entropy $S_k$ for $k = -12, \ldots, 12$ between the 1 month and 6 month LLEs (Iquique–SST) and 1 month LLEs (Arica–AST) and the ONI index. Left column: original time series. Right column: pre-whitened time series. Iquique 1 month and 6 months corresponds to the first two upper graphs and the two central ones; while Arica 1 month corresponds to the lower two graphs.

through hydroclimatological variables such as wind and temperature. Its eastern phases appear to be alternately related to the warm component of ENSO (Riveros et al., 2020). The QBO has already been seen in other records from the Chilean region, between 39° and 43° S (Villalba et al., 1998). Fagel et al. (2008), when analyzing layered sediments from Lake Peyuhue, 40°

S, 72° W, found that the frequency bands of the thickness of the layers were mainly related to QBO periodicities. In our case, we found this oscillation in oceanic and atmospheric surface temperature records. For the oceanic records it was found in all the stations, something that did not happen with the atmospheric records, where it is only seen for the stations in the central and southern parts of Chile. Similarly, periods related to ENSO and PDO were obtained, as well as some others that we believe may be related to local events, however, it should be noted that QBO has also been related to ENSO.





Regarding the periodicity of 1.5 years found in both classes of station, we have not been able to associate this with any internal or external forcing.

## 4.2   Nonlinear Analysis and ONI index

The effects of changes in the SST and AST on biota range from the redistribution of species (Lubchenco et al., 1993; Ihlow et al., 2015), to the increase in vulnerability as occurs in the HCS due to the expansion of oxygen minima associated with

the increase in SST (Mayol et al., 2012; Stramma et al., 2008), to increases in extreme events (Harris et al., 2018; Garreaud, 2018), up to subtle changes in biological–climatic feedbacks (Bradford et al., 2019; Bardgett et al., 2008), all of which invite discussions about the contribution of the oceans in these feedbacks (Joos, 2015). For these reasons, understanding the effects of climate variabilities such as El Niño on the behaviour of long time series of coastal temperature records can contribute to improving forecasts and understanding the effects of climate change and climate variability in systems such as the HCS (Harris

et al., 2018).

If we divide the region into northern, central, and southern segments in our wide study area of 18°–45° S based on the latitudinal climatic gradient, we can offer the following bioclimatic interpretations. The results point to a globally non–chaotic dynamics for both oceanic and atmospheric stations. LLE analysis highlights patterns of reduced and enhanced stability and predictability in the phase space. As expected, the LLEs approach the global negative Lyapunov exponent as the number of steps

ahead increases, but the asymptotic behavior of the LLEs towards the global exponent differs in oceanic versus atmospheric stations; stability is reached faster in oceanic stations. This may be expected, since the oceanic environment is less perturbable than the atmosphere owing to the large heat capacity of water.

In both oceanic and atmospheric stations, Antofagasta shows the slowest speed of convergence; in the oceanic case, accompanied by Corral. Antofagasta is characterized by the constant presence of warm water inside its bay (Piñones et al., 2007), so

its climate possesses a type of local permanent El Niño, keeping the bay in a state of external forcing. This makes the dynamics of the Antofagasta system look unstable in the LLEs, and when comparing with ONI it is not possible to distinguish different behavior when a general El Niño event occurs. Regarding Corral, there is a cooling trend in the coastal upwelling regime along northern Chile and southern Peru, which contrasts with the warming trend in the last 350 years in an offshore area of the Humboldt Current system at 36° S (Vargas et al., 2007). Such El Niño–type conditions found also in Corral might explain its

reduced speed of convergence.

The climatic effects of El Niño, as well as the forcings induced by it, are part of a complex system, since interactions between different factors are necessary for certain processes to occur, such as primary production and variability in the region's climate. A linear analysis cannot decipher the behavior of the climate system in any region; likewise, the results of linear methods are usually ambiguous and inadequate, since natural processes are complex, show non-stationarity and are recorded mainly as

short and noisy series (Marwan et al., 2003). The relationship between climate forcing and temperature change is not expected to be linear in this type of data, owing to the nonlinear relationships between coastal upwelling, the interdecadal variability of El Niño and global warming in the Chile/Peru margin (Vargas et al., 2007). The regional effects of El Niño in this arid zone interfere with the effects caused by the Humboldt Current, not only within it, but also in the entire Pacific subtropical



anticyclone, since during El Niño conditions it undergoes a large–scale weakening (Rutllant et al., 2003). This decrease is seen
to be gradual, as the behavior of the stations changes considerably with the different windows.

The case of Iquique–SST is peculiar due not only to its attractor, which shows an abrupt change and then returns to the same
behavior, but also because, of all the oceanic stations in terms of neural network analysis, it is the one that requires a lower
embedding dimension ($d = 3$) than the rest of the stations, likewise, it is the only station that presents mixed behavior very
early, that is, in the 1 month window. Also, it shows a positive correlation between the ONI index and LLEs. This indicates a
strong relationship between the climatic variability of this region and the El Niño phenomenon. Fuenzalida (1985), through an
analysis for El Niño 1982–83, showed that an increase in temperature in Iquique is associated with this phenomenon. In our
case, the cross–correlation test yielded a significant correlation at lag $+1$ for the 1 month time horizon and lag $-2$ for the 6-
month time horizon. The difference between the two is that the record observed at 1 month presents stabilities and instabilities
which, together with other factors discussed below, favor a forward correlation and could be associated with the warm phase
of ENSO. For the 6 month time horizon, the behavior of the area is completely stable; perhaps for this reason there is a lagging
effect associated with the cold phase of ENSO.

Notably, like La Serena and Valdivia, where a similar dynamics occurs between them – discussed below —, Iquique is also
in a transition zone, the faunal transition zone, located south of the subtropical and north of the subantarctic zone (Bé and
Tolderlund, 1971), where it is possible to observe a mixture of planktonic species formed by foraminifera from both cold and
warm waters. In addition, the northern part of Chile, in which Iquique is located, is an area with wind-driven coastal upwelling
throughout the year. Barbieri et al. (1995) analyzed the northern region and found four sources of upwelling in Iquique. The
upwelling south of Iquique is the most important due to its latitudinal extension. For this area the highest correlation between
wind and SST is observed.

It is due to all these oceanic and environmental characteristics that exist in Iquique, that we hypothesize that the abrupt
change observed in the trajectory of the SST transition could be due to the climate of 1976–77, a coupled global variation of
the atmosphere–ocean system that links the variability of the sea surface temperature of the Pacific and climatic parameters in
most of the world (Namias, 1978; Houghton et al., 2001; Solomon et al., 2007). In addition to the strong influence of ENSO on
the interannual time scale, the most notable feature is the abrupt upward shift in air and ocean temperature regimes in the mid
1970s. Although this was a global variation, probably due to all the aforementioned characteristics, it is possible in Iquique
to observe this abrupt change in its trajectory, something that does not happen with the other stations, where the shape of the
attractors indicates the presence of nonlinear oscillators, possibly forced. On the other hand, it is these same characteristics that
probably make Iquique's dynamics more consistent with ENSO events and favor that its stability/instability characteristics are
different from the rest of the oceanic stations.

In the same way, we found similarity between the dynamics of Arica — upwelling throughout the year — and Talcahuano
— seasonal upwelling. This could be the result of large-scale oceanic processes that occur simultaneously in both places. In
Castro et al. (2020), similar variations of $\delta^{13}C$ of particulate organic matter were noted in Iquique and Talcahuano, despite
being two totally different areas; this was hypothesized to be influenced by coastal outcrop. In our case, since Arica is not very
far from Iquique, we can think of this outcrop as part of the explanation for the similarity of the dynamics that we found. On



the other hand, we have the northward currents that carry subantarctic water, as well as the intrusion of subtropical waters to
the south, which during El Niño conditions can approach the coastal zone, the propagation of trapped waves to the coast and a
poleward subsurface flow. From a dynamical point of view, this variability can affect currents, the mixing of the water column,
the intensity of upwelling, as well as the temperature and sea level on the continental shelf and slope throughout the Humboldt
Current system (Shaffer et al., 1997, 1999). In addition to this, the Chilean continental shelf is extremely narrow compared to
the Atlantic coast, with a maximum width close to 45 km in the Talcahuano area and a maximum depth in general of 150 m,
except for the Valparaíso area, where it reaches up to 800 m (Camus, 2001). So, these may be other possible explanations for
the similarity in dynamics found in these two regions. However, something else must be happening between the two stations
to maintain this dynamics in the three study windows.

   In the atmospheric stations it is also possible to observe similar dynamics between Rodelillo, Puerto Montt and Antofagasta–
AST, with Antofagasta–SST. In these three stations we can observe repetitive and persistent behavior throughout the registry
and, again, we are talking about the northern and southern regions of Chile. Therefore, there are probably teleconnection
processes throughout the zone. In this respect, Gómez (2008) made hydrographic and meteorological observations during the
period 1996–2005, finding for the central-southern zone of Chile, an interannual and decadal modulation of the beginning of
the upwelling season during the period 1968–2005. This modulation would be linked to changes in atmospheric circulation
patterns, associated with teleconnections triggered during El Niño. For this study, it is also necessary to add the presence of
teleconnections in oceanographic processes. Finally, for the atmospheric stations of Arica, Iquique and Antofagasta, a decrease
in unstable behavior can be seen during El Niño events, possibly due to the effects of the Pacific anticyclone. Regarding Arica,
the cross entropy test showed a significant correlation between ENSO and its LLE for the 1 month time horizon, showing two
lag values, one with lag $+2$ and the other with lag $-3$. The values are larger than in Iquique; however, Iquique is oceanic, while
Arica is an atmospheric station and the latter is easier to perturb, which could cause these values.
The oceanic and atmospheric stations not only are arranged latitudinally, but one forms the counterpart of the other, that
is, the oceanic part has its atmospheric counterpart, for example, Antofagasta–SST with Antofagasta–AST, Iquique–SST with
Iquique–AST, Coquimbo–SST with Serena–AST, and so on. This is important since when observing the behavior of the stations
in the different windows, it was noticeable that for the three windows the behavior of Corral and Valdivia, its counterpart, were
similar.
For the AST stations, it can be noted by eye that Rodelillo behaves differently from the rest of the series. For the 6 month
window it has more stable behavior, most noticeably in the first part of the series, than the rest of the stations, and for the 1
year window it is the only one that behaves in a stable manner in its entirety.

   Puerto Montt is the southernmost station, on the northern border of Patagonia (Garreaud et al., 2013). This station is located
well inside the continent, on the coast of an inland sea, with greater thermal amplitudes due to the continental effect. As this
station is located south of the southern limit recognized by the influence of El Niño, it is possible to assume that El Niño
is not responsible for the climate variability that may occur in this station, since we can see an unstable dynamics also in
the oceanic part (Puerto Montt–SST), but rather Southern Annual Mode (SAM) (González-Reyes, 2013). However, ENSO
periods can generate waves capable of interacting with the region (Karstensen and Ulloa, 2009), thus contributing to unstable





dynamics. However, for this region there is already a high degree of disturbance, so what could be happening here are further
intensifications induced by El Niño. This may be the reason why a behavior similar to that of Antofagasta–SST is also observed
when we compare the ONI index and the LLE–AST of both stations.

In the case of Rodelillo, we can hypothesize that the local climate is greatly influencing the results obtained; this series
was divided into two parts, which showed homogeneous results for the three different types of windows, that is, if the first
segment of Rodelillo for the 1 year window was stable, the second was, too. This area is in the central zone of Chile and
both the central and central-southern zones of Chile (32°–43° S) concentrate most of the wildfire activity in the country (Soto
et al., 2015; Urrutia-Jalabert et al., 2018). Data from Rodelillo Meteorological Station located in the Valparaiso region reported
daily maximum temperatures and thresholds (90 %) of heat waves in Chile during 2017, 2018, and 2019 (Guerrero et al.,
2021). It has been shown that phenomena such as ENSO and the Antarctic Oscillation AAO have a significant influence on
fire dynamics — anomalies in rainfall occur during the year prior to the fire season (Urrutia-Jalabert et al., 2018) — and on
vegetation, primarily through their influence on regional climate and fuel conditions (Swetnam and Betancourt, 1998, 1990;
Kitzberger and Veblen, 1997; Veblen et al., 2000; Holz et al., 2012; Mariani et al., 2016). Fire history studies over several
centuries have linked wildfire activity in southern Chile and Argentina to hot, dry conditions associated with the positive phase
of the AAO, as well as to below-average precipitation conditions associated with negative deviations of ENSO and the Pacific
Decadal Oscillation (Holz and Veblen, 2012; Mundo et al., 2013; Holz et al., 2017).

For their part, La Serena and Valdivia show a similar dynamics for the 6 months windows. La Serena, in a transition zone
between the hyperarid northern Atacama Desert and the more mesic Mediterranean region of southern central Chile, is strongly
affected by ENSO, which is the primary driver underlying the high interannual variability in precipitation, and which promotes
higher average rainfall during the warm phase (Rutllant and Fuenzalida, 1991). On the other hand, Valdivia is located within
a transitional area between the influence of ENSO and SAM. This area, the Valdivian ecoregion, is considered a site of high
biodiversity and importance for biological conservation. The dynamics of both seasons are influenced by ENSO; however,
the effect of the Tropical South Pacific Anticyclone also plays an important role in this similarity, as the effects of this semi-
permanent high pressure system change from 35° S, 90° W in January to 25° S, 90° W in July (Kalthoff et al., 2002; Montecinos
et al., 2015), that is, in a period of 6 months, which is one of the windows that we use and in which important changes in the
dynamics of the systems can be seen. The influence of the anticyclone is permanent to the north of 30° S, maintaining dry
conditions, and weak south of 40° S (Camus, 2001), where there is rain throughout the year, which is why it has important
effects in Valdivia. Mohtadi et al. (2005) and Marchant et al. (2007) recognized two areas of high productivity on the Chilean
coast: 1) in the northern area (24° S and between 30°–33° S) and 2) south of 39° S. These two areas encompass both La Serena
and Valdivia, thus indicating the important contribution of the anticyclone to climate variability in Chile.

In the cross–entropy test for the time horizons of 1 month and 1 year performed at La Serena, Fig. A17,where important
correspondences between ONI and LLE could be observed, it was possible to verify that there is no significant correlation.
This indicates that, although the instability of the dynamics in Serena is influenced by ENSO, it is not this phenomenon that
contributes the most to the instabilities. The same is true with Valdivia, where the test did not indicate a significant correlation,





either. This was to be expected because this area is also influenced by SAM and it is likely that the latter has a greater influence on the area.

Only Iquique–SST, for time horizons of 1 and 6 months, presented a significant correlation with ENSO, while for the atmospheric stations, it was Arica with a time horizon of 1 month that showed a significant correlation. That is, it is only for the northern part of Chile that we obtained correlations between the dynamics of climate variability in Chile seen through the LLE and the ENSO. A possible explanation for this correlation found in both places may be the disturbances coming from the equator that only influence the northern part of Chile and that can affect the physical environment (Pizarro et al., 1994).

This favors the circumstances for there to be a correlation between ENSO and the LLE in this area; on the other hand, the initial conditions are important for the influence of westerly winds: during the beginning of a warm event, a westerly wind can accelerate the development of the event, while one after the peak of El Niño will simply prolong its duration (Fedorov, 2002; Fedorov et al., 2003). The majority of the published studies on ENSO in Chile report findings for the northern and central regions (Gutiérrez and Meserve, 2003; Holmgren et al., 2001; Guera and Portflitt, 1991; Vilina et al., 2002; Ulloa et al., 2001;

Acosta-Jamett et al., 2016); although ENSO is a phenomenon that affects the climate globally, it has a greater influence on the northern part of our study region.

Each El Niño event will affect different areas of the Chilean region differently, since each one is unique. This can be seen in the study by Barbieri et al. (1995), who observed that during the 1987 El Niño the mean SST showed values above 1.5 °C throughout the year, while during the 1992 El Niño these anomalies occurred from March to May and from September

to November. It is a fact that El Niño influences the regional climate, as well as the PDO and the SAM. That is why for future studies it is proposed to analyze in greater detail the influence of the last two, as well as the level of impact that regional characteristics have, in such a way that great improvements are achieved in the predictability of these climatological phenomena.

## 5 Conclusions

We analyzed with a LLE analysis the patterns of local instability for the 16 stations, and these correlate on some occasions with ENSO events. However, it is only for the northern stations, specifically Iquique–SST and Arica–AST that a significant correlation can be observed. ENSO, despite being a phenomenon that affects the climate globally, has greater influence in the northern part of Chile.

Teleconnections are noted throughout the entire region that may be associated with ENSO, PDO, SAM, the Pacific Anti-

cyclone, upwelling zones and QBO. A link can be seen between the climate variability of the region and different internal forcings (PDO, ENSO, SAM, Pacific Anticyclone) and external ones (QBO, Lunar Cycle) that contribute to the complexity of the system and favor a change in the variability as we move in latitude, which is due, in part, to the fact that it is necessary to take regional characteristics into account if one wants to understand the response of the different study systems, oceanic and atmospheric, to the different forcings.





Previously unused Chilean naval temperature data from a large latitudinal spread of stations have helped us to uncover a great variety of factors involved in the temperature dynamics of the Chilean region. We find a non–chaotic climate variability, but with a nonlinear behaviour; reinforcing what has been mentioned in the literature. Our study invites more detailed work on the northern part of Chile, especially on Iquique and its sea surface temperature, where the behavior of the system was very peculiar both in its low embedding dimension, as well as in the shape of the attractor and the significant correlation with ENSO

for time horizons of 1 and 6 months.

*Data availability.*   We obtained access to Chilean naval temperature data; the rights to those data remain with the navy.

## Appendix A:  A

### A1    he Humboldt Current system, El Niño–Southern Oscillation, and Southern Annual Mode

The Humboldt Current System (HCS) that brings water from the tropical and sub–polar regions is characterized by evident

interannual and seasonal variabilities (Mayol et al., 2012). The system is composed of: (i) the Humboldt Current predominantly away from the equatorial coast that moves at an average surface velocity of 6 cm s$^{-1}$; (ii) coastal currents formed by the Peru–Chile current and countercurrent; and (iii) the Peru–Chile equatorial coastal current. The currents towards the pole are responsible for transporting equatorial and subtropical subsurface waters to the Chilean coasts, while the flow towards the equator brings cold Antarctic and Antarctic intermediate waters. The HCS is controlled to a large extent by the equatorial

coastal winds linked to the Pacific subtropical anticyclone, which promotes the coastal upwelling associated with the primary productivity in the north and center of Chile, extending its influence to the south of Chile in summer (Montecino et al., 2006). Around the western tropical zone of the Pacific there are ocean–atmosphere interactions that have effects at the planetary level (Enfield and Allen, 1980). In particular there is the El Niño–Southern Oscillation (ENSO) phenomenon, characterized by two opposite phases: one of warming and rainfall in the Eastern Pacific, known as El Niño, and a second of cooling and dry years

called La Niña (Pizarro and Montecinos, 2004). The redistribution of temperature in the water column can be associated with the passage of low-frequency coastal trapped waves (CTW) (days at the intra–seasonal scale) and cool and warm interannual anomalies produced by equatorial waves due to the Southern Oscillation of El Niño (Shaffer et al., 1997; Montecinos and Gomez, 2010). In addition to ENSO, there is the Pacific Decadal Oscillation (PDO), which develops in the northern Pacific Ocean. It may be described as a long period ENSO (Núñez et al., 2013), since it also has a positive or warm phase and a

negative or cold phase, but a different time scale. The PDO can intensify or diminish the impact of ENSO, depending on the relative phase in which these two oscillations are found. When ENOS and PDO are in phase, the impacts induced by El Niño or La Niña will be magnified with respect to normal patterns. Conversely, if ENSO and PDO are in opposite phases, the effects on global climate variability will be weaker (Wang et al., 2014). The Southern Annual Mode (SAM) or Antarctic Oscillation defines the changes in the westerly winds that are driven by atmospheric pressure contrasts, which in turn generate pressure

differences between the tropics and the southern polar areas. The change of position of the western wind band, produced from





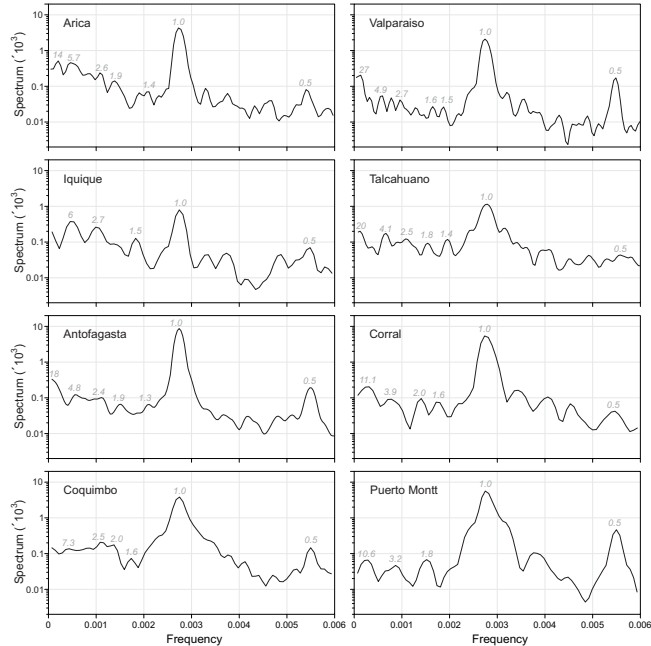

**Figure A1.** Spectral analyses for oceanic stations. The numbers indicate the spectral peaks with the highest energy.

west to east in latitudes between 30° and 60° of both hemispheres, influences the strength and position of cold fronts and mid–latitude storm systems. In the positive phases of SAM, a belt of strong westerly winds contracts towards Antarctica. This results in weaker than normal westerly winds and high pressures in southern Australia, restricting the entry of colds fronts. SAM differentially affects the surface temperature of the four continental masses of the Southern Hemisphere, including part

of the area influenced by the HCS at mid–latitudes (Gillett et al., 2006). In conjunction with ENSO, SAM also explains part of the coastal temperature anomalies at high (Fogt and Bromwich, 2006), and also at mid latitudes (Yeo and Kim, 1992).

## A2 Spectral Analysis

Figures A1 and A2 show spectral analyses performed for the oceanic and atmospheric stations. Numbers are given for the spectral peaks with the highest energies.





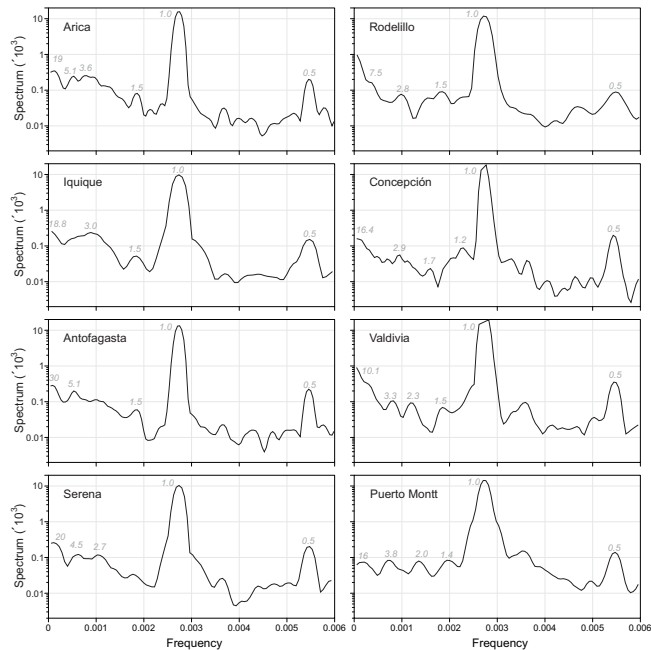

**Figure A2.** Spectral analyses for atmospheric stations. The numbers indicate the spectral peaks with the highest energy.

**A3 Trend Estimation**

In Figures A3 and A4 trend estimations are given for the 16 stations in a latitudinal manner, north to south.





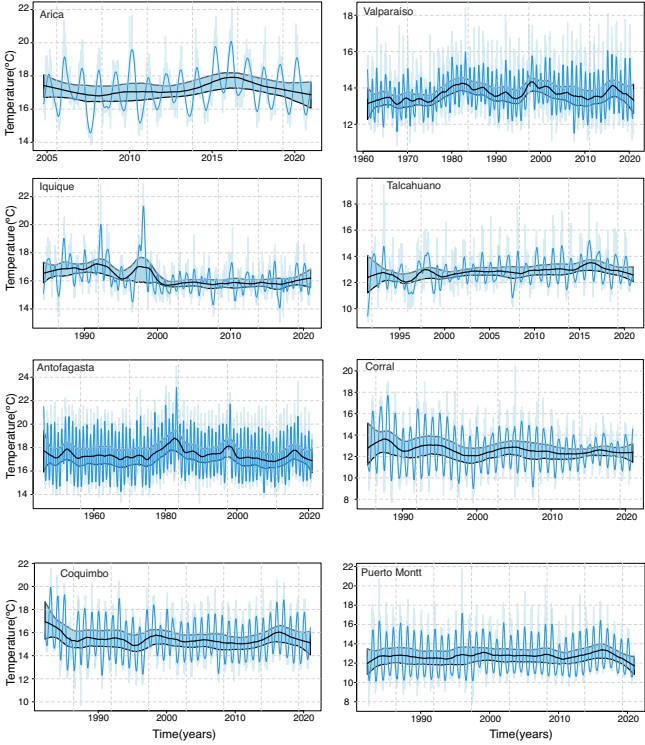

**Figure A3.** Trend estimation for oceanic stations. Interpolated original (light blue), trend–cycle (blue) and long–term trend (black), together with confidence bands at 95 % (light blue).





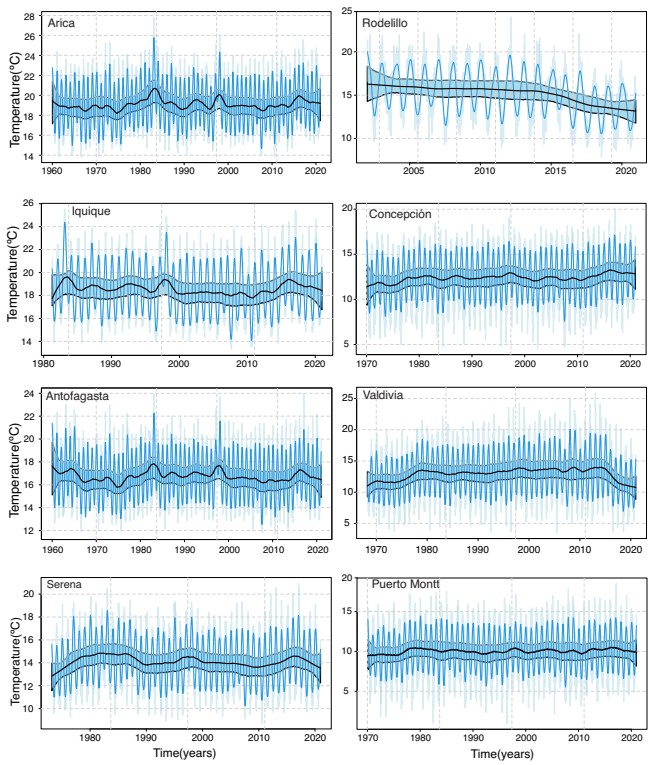

**Figure A4.** Trend estimation for atmospheric stations. Interpolated original (light blue), trend–cycle (blue) and long–term trend (black), together with confidence bands at 95 % (light blue).

## A4  Local Lyapunov Exponents

In Figures A5 to A10 local Lyapunov exponents are shown for all 16 stations for 1 month, 6 month and 1 year windows.





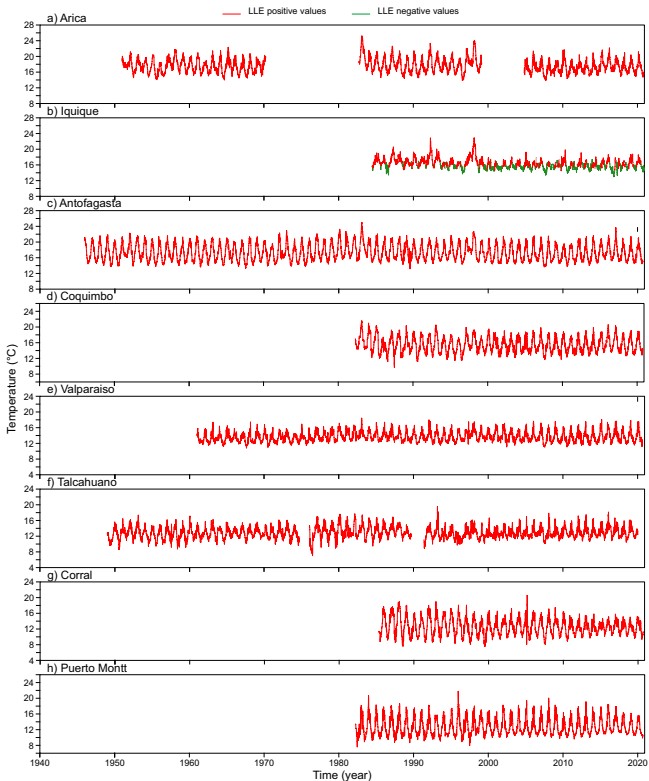

**Figure A5.** Temperature record for ocean stations with superimposed information on 1 month local Lyapunov exponents: positive values for the LLEs are shown in red, negative values are shown in green. Stations are given in a latitudinal arrangement.





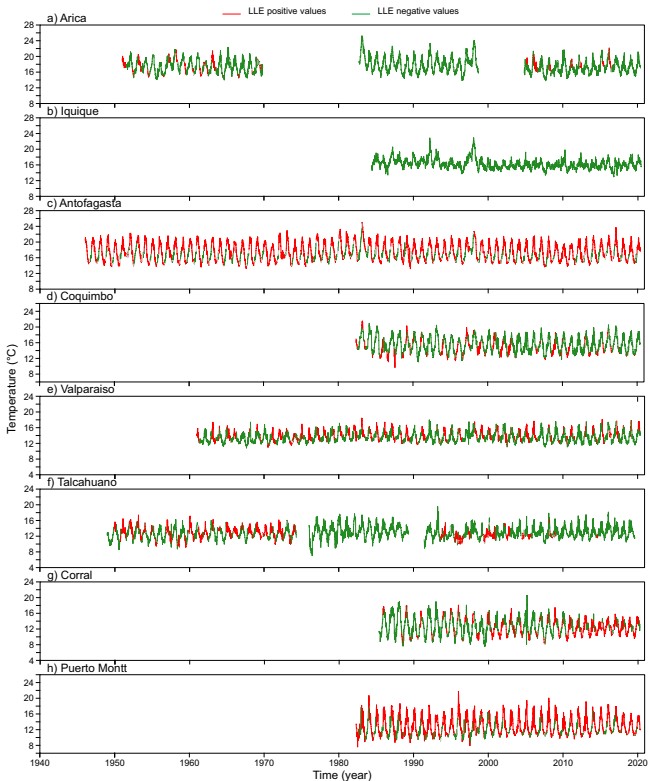

**Figure A6.** Temperature record for ocean stations with superimposed information on 6 month local Lyapunov exponents: positive values for the LLEs are shown in red, negative values are shown in green. Stations are given in a latitudinal arrangement.





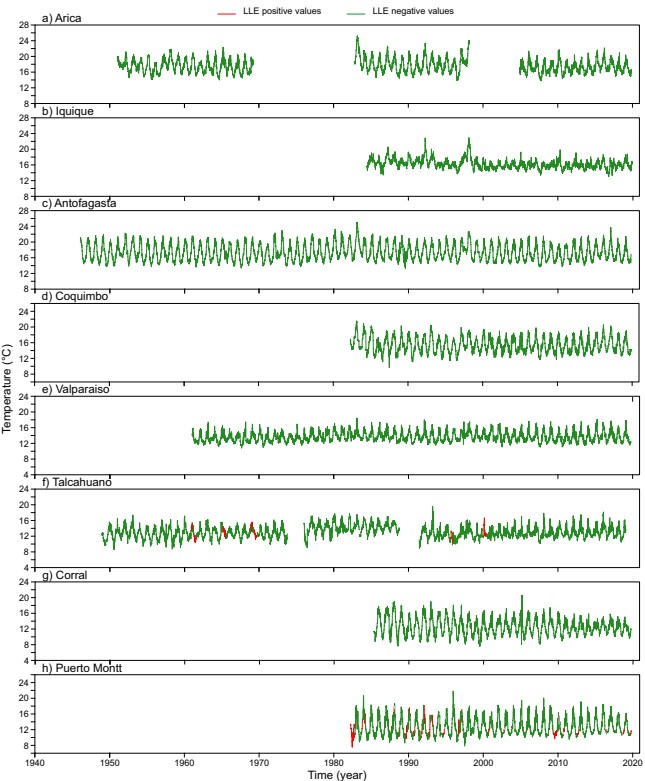

**Figure A7.** Temperature record for ocean stations with superimposed information on 1 year local Lyapunov exponents: positive values for the LLEs are shown in red, negative values are shown in green. Stations are given in a latitudinal arrangement.





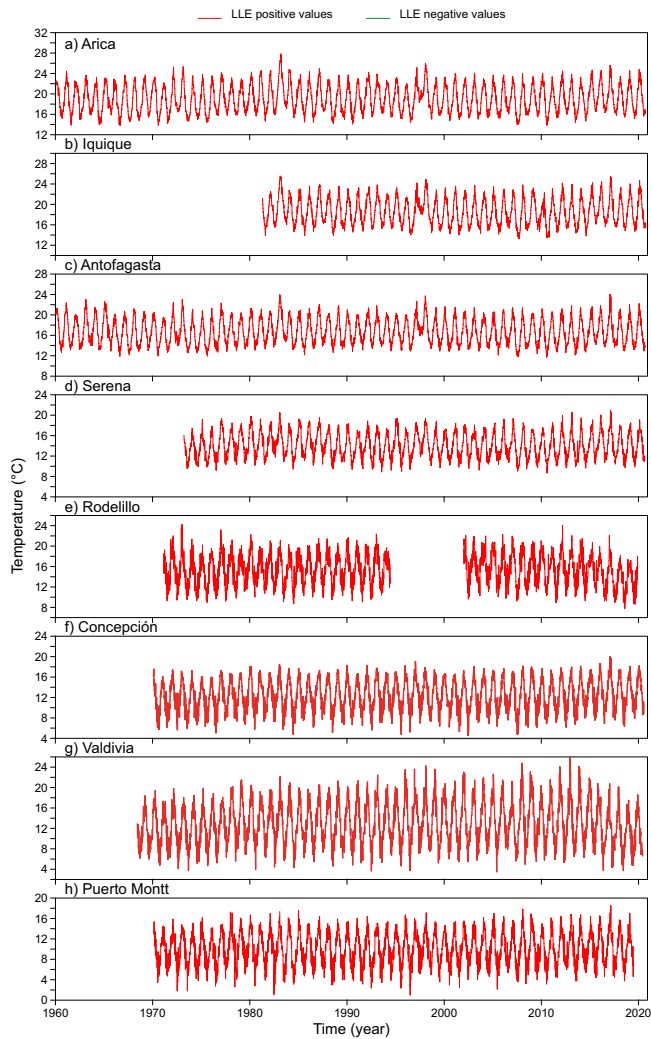

**Figure A8.** Temperature record for atmospheric stations with superimposed information on 1 month local Lyapunov exponents: positive values for the LLEs are shown in red, negative values are shown in green. Stations are given in a latitudinal arrangement.





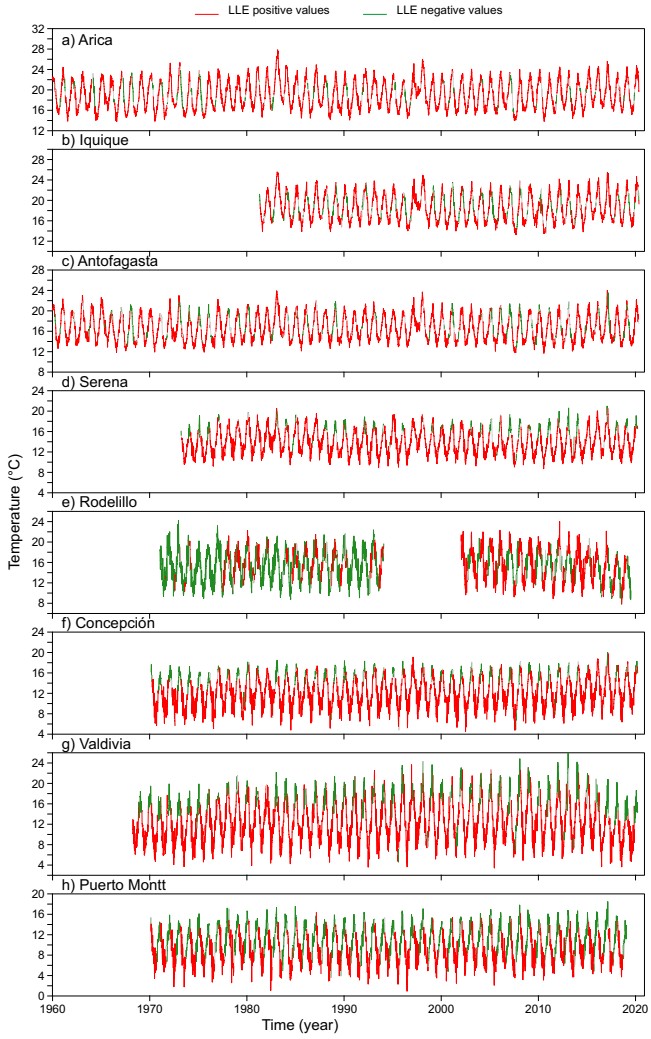

**Figure A9.** Temperature record for atmospheric stations with superimposed information on 6 month local Lyapunov exponents: positive values for the LLEs are shown in red, negative values are shown in green. Stations are given in a latitudinal arrangement.





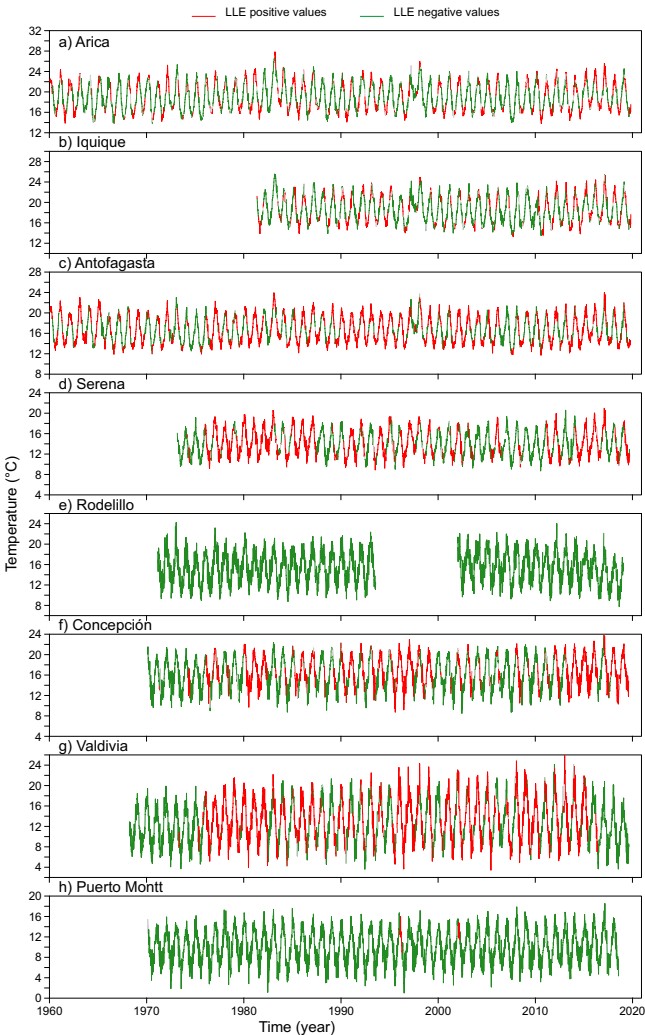

**Figure A10.** Temperature record for atmospheric stations with superimposed information on 1 year local Lyapunov exponents: positive values for the LLEs are shown in red, negative values are shown in green. Stations are given in a latitudinal arrangement.

## A5   Cross entropy $S_k$

Let $\{X_t\}$ and $\{Y_t\}$, $t \in \mathbb{N}$, be two stationary random processes, where $F_{X_t,Y_t}(x,y) = P(X_t \leq x, Y_t \leq y)$, $F_{X_t}(x) = P(X_t \leq x)$, $F_{Y_t}(y) = P(Y_t \leq y)$. Then, the metric entropy $S_\rho$ at lag $k$ is a normalized version of the Bhattacharya–Hellinger–Matusita distance, defined as

$$S_\rho(k) = \frac{1}{2} \int \int \left( \sqrt{dF_{(X_t,Y_{t+k})}(x,y)} - \sqrt{dF_{X_t}(x)\,dF_{Y_{t+k}}(y)} \right)^2, \tag{A0}$$

$$= 1 - \int \int \sqrt{dF_{(X_t,Y_{t+k})}(x,y)\,dF_{X_t}(x)\,dF_{Y_{t+k}}(y)}. \tag{A0}$$





In the case where $Y_t = X_t$ for all $t$, $S_\rho$ measures the serial dependence of $\{X_t\}$ at lag $k$ and can be interpreted as a nonlinear auto/cross-correlation function that overcomes the limits of Pearson's correlation coefficient. As pointed out in Maasoumi (1993); Granger et al. (2004), and Giannerini et al. (2015), $S_\rho$ satisfies many desirable properties, including the seven Rényi axioms and the additional properties described in Maasoumi (1993). In the following, we use $S_k$ instead of $S_\rho$ for simplicity. We compute the cross entropy as detailed in Giannerini and Goracci (2023). In Figures A11–A16 we show the results of

the cross-entropy test between the ONI and LLEs for the oceanic stations with a time horizon of 1 and 6 months, and for the atmospheric stations with a time horizon of 1 month. For Serena we also tested the 1 year–ahead LLEs, see Fig. A17. For the atmospheric stations, only 1 month LLEs are shown, since no relationship was found between the ONI index and 6 month–ahead LLEs.



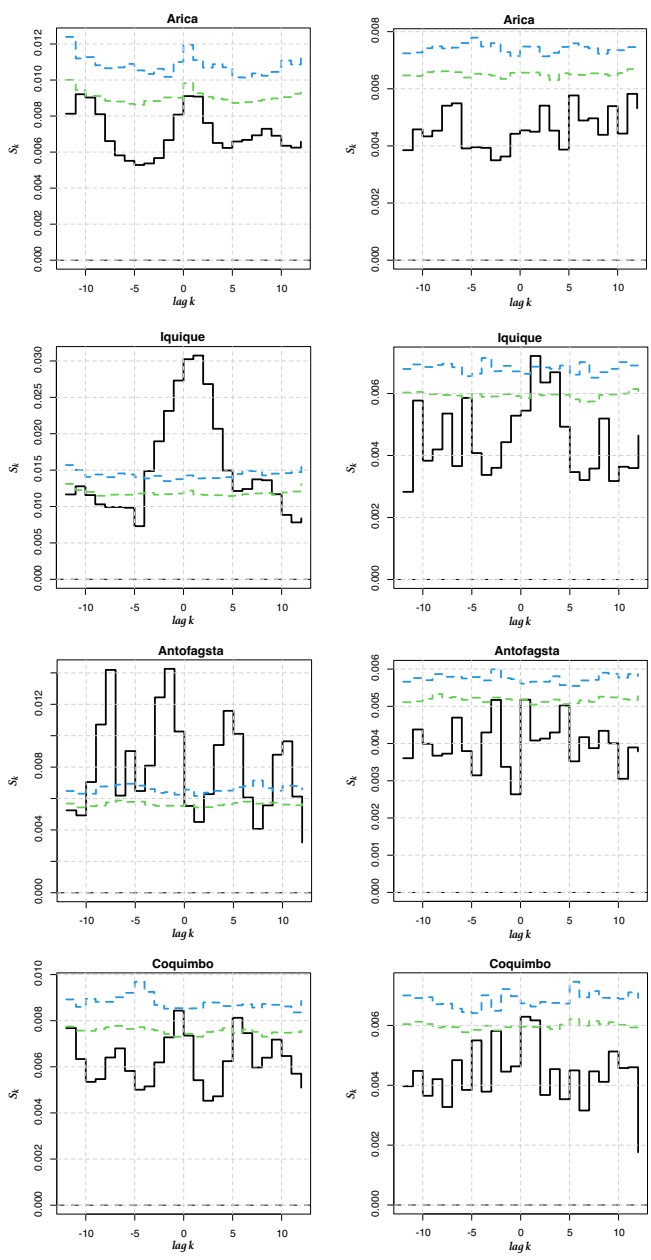

**Figure A11.** Cross entropy $S_k$ for $k = -12, \ldots, 12$ between the 1 month LLEs (SST) and the ONI index for the first 4 stations. Left: original time series. Right: pre–whitened time series.

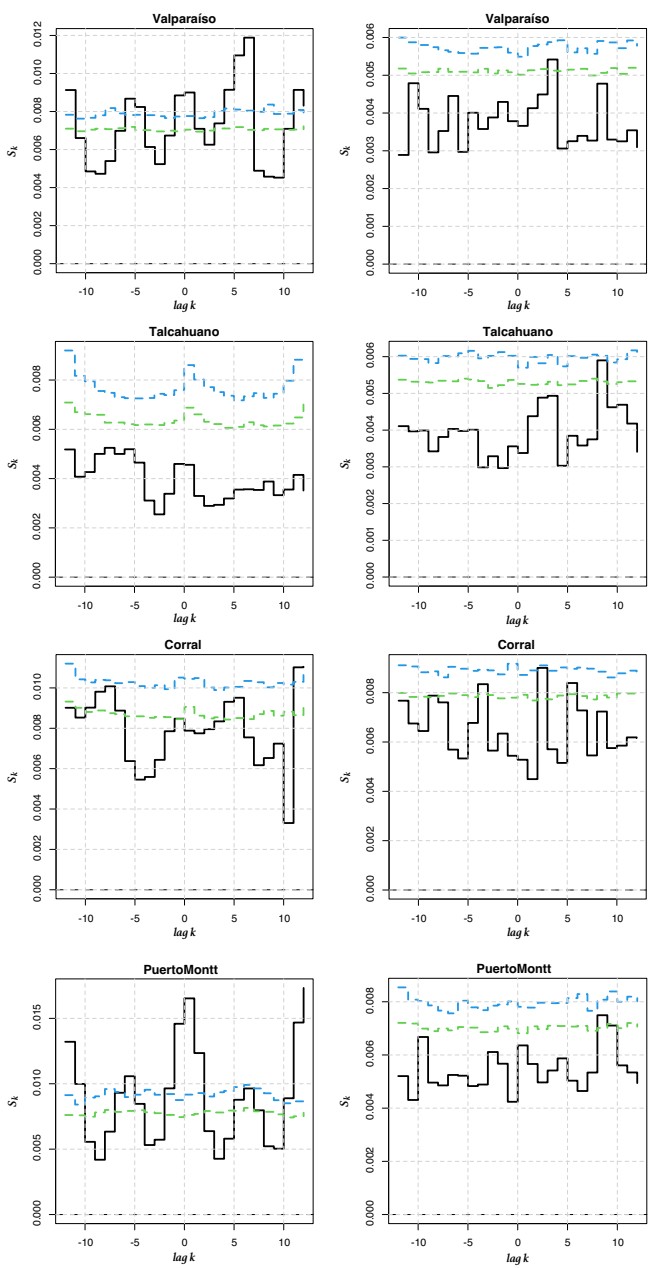

**Figure A12.** Cross entropy $S_k$ for $k = -12, \ldots, 12$ between the 1 month LLEs (SST) and the ONI index or the last 4 stations. Left: original time series. Right: pre–whitened time series.





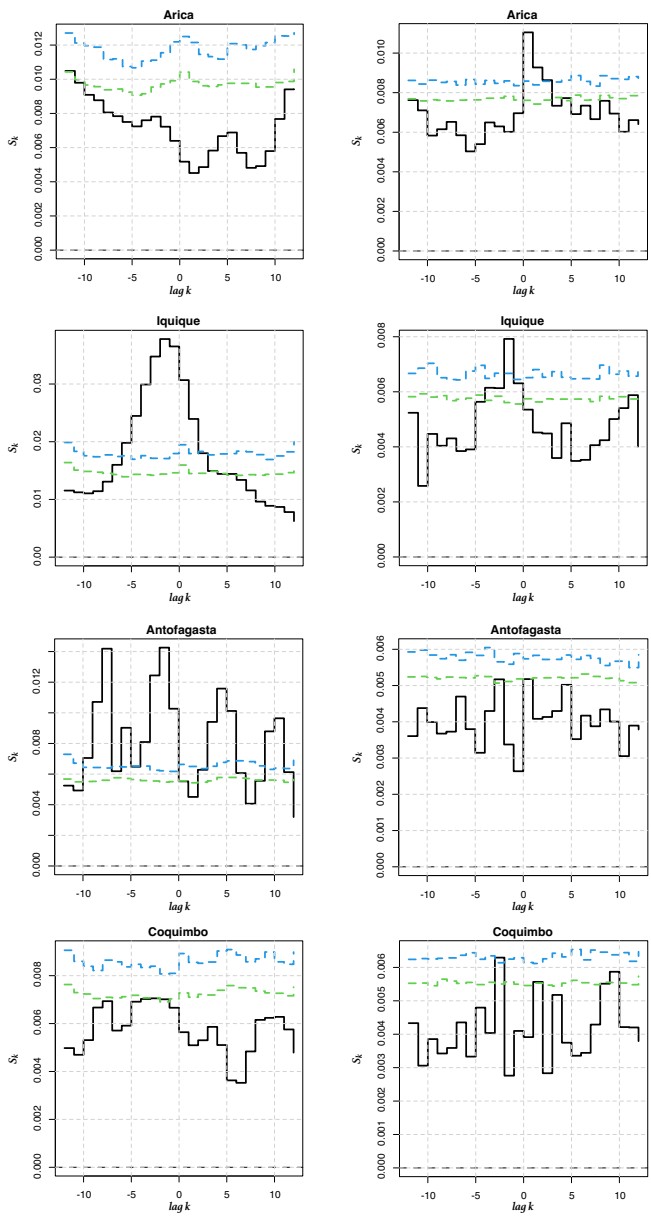

**Figure A13.** Cross entropy $S_k$ for $k = -12, \ldots, 12$ between the 6 months LLEs (SST) and the ONI index for the first 4 stations. Left: original time series. Right: pre–whitened time series.





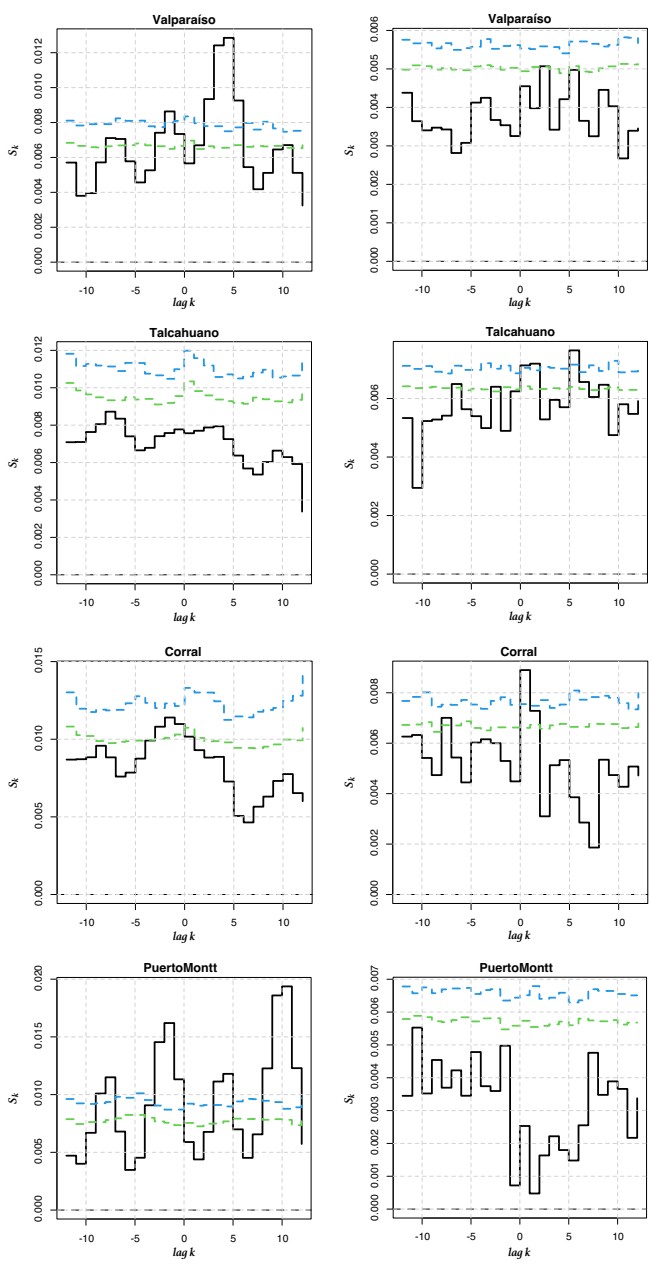

**Figure A14.** Cross entropy $S_k$ for $k = -12, \ldots, 12$ between the 6 months LLEs (SST) and the ONI index for the last 4 stations. Left: original time series. Right: pre–whitened time series.



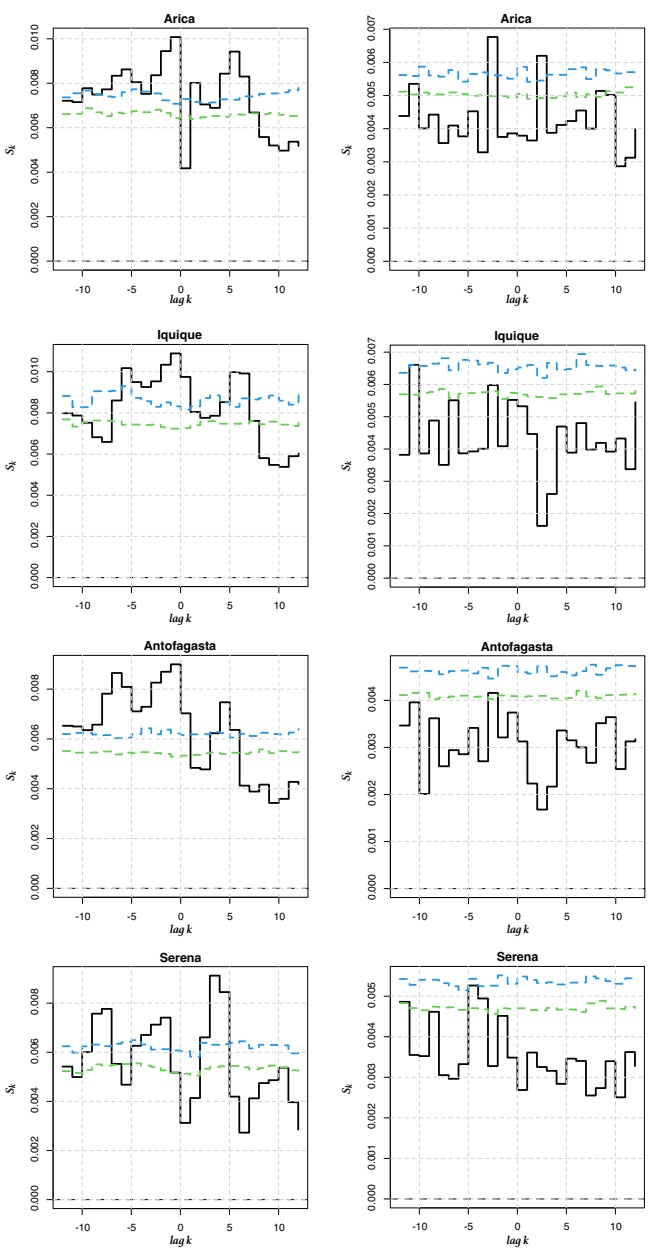

**Figure A15.** Cross entropy $S_k$ for $k = -12, \ldots, 12$ between the 1 month LLEs (AST) and the ONI index for the first 4 stations. Left: original time series. Right: pre–whitened time series.



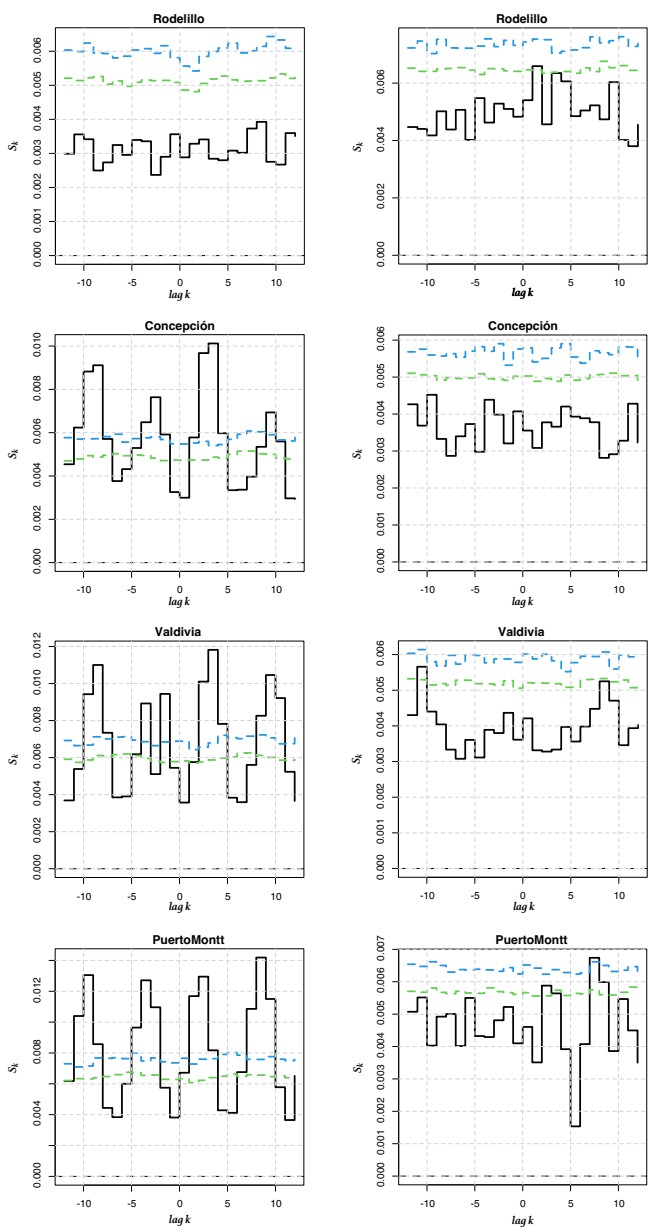

**Figure A16.** Cross entropy $S_k$ for $k = -12, \ldots, 12$ between the 1 month LLEs (AST) and the ONI index for the last 4 stations. Left: original time series. Right: pre–whitened time series.





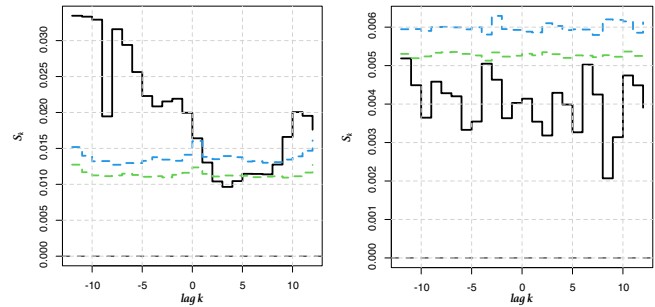

**Figure A17.** Cross entropy $S_k$ for $k = -12, \ldots, 12$ between the 1 year LLEs for La Serena and the ONI index. Left: original time series. Right: pre–whitened time series.

*Author contributions.* Berenice Rojo-Garibaldi, Simone Giannerini and Julyan Cartwright participated in the development, design, calculations, elaboration of figures and the first draft of the article. David Alberto Salas-de-León, Verónica Vázquez-Guerra and Berenice Rojo-Garibaldi participated in the elaboration of the results' discussion. Manuel Contreras provided all the data and helped with the development of the first ideas. All authors read and approved the final manuscript.

*Competing interests.* All authors declare that they have no conflict of interest.

*Acknowledgements.* Berenice Rojo-Garibaldi would like to thank Dr. Thomas Seligman of the Institute of Physical Sciences, UNAM, Campus Cuernavaca, Mexico, for his helpful comments during the course of this work.





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
