# Peer review of "Nonlinear Time Series Analysis of Coastal Temperatures and El Niño-Southern Oscillation Events in the Eastern South Pacific"

_Earth System Dynamics, 2023_

## Author Comment (AC1)

**Response to Referee 1**

Rojo-Garibladi, Contreras–López; Giannerini; Salas–de–León; Vázquez–Guerra; Cartwright

August 21, 2023

Dear Referee,

many thanks for your comments, please find below a pointwise reply to all of them, where your comments are in italic.

1. *In the introduction authors ought to add some considerations on the impact of El Niño on human life, and, in particular, on human health. This would enhance the relevance of their work and increase its readership.*

   We propose a modification to the paragraph, adding the following:

   In a context of global warming, it is important to understand the spatial and temporal patterns of atmospheric temperatures in complex systems like the Eastern South Pacific region. Gaining knowledge on temperature changes, and in the phenomena behind them, is essential for understanding both scientific and societal issues, since El Niño events have a great impact on human life [Glantz, 2022], and, in particular, on human health [Kovats et al., 2003] far beyond the Eastern South Pacific region [Fan et al., 2017].

2. *Section 2.1.2 needs more bibliography.*

   thank you, we have added more bibliography.

3. *Section 2.1.2: the final sentence on Friedrich et al must be anticipated and also expanded: it is not good to say the readers: go and read that paper. Moreover, the journal is on a topic that is very different from the one of this journal, thus maybe readers could incur in some difficulties in reading that paper. For example, in section 2.1.3 authors succintlu but clearly describe the main content of various papers*

   Thank you for the comment, we propose to add the following at the beginning of Section 2.1.2:

   > One recurring issue with trend estimation is that, in most situations, the detrended series is both dependent and possibly heteroskedastic; moreover, missing data are very common and disregarding these aspects leads to invalid confidence bands. Here, we follow Friedrich et al. [2020], that solve the problem by proposing a novel autoregressive wild bootstrap scheme that does not need adjustments in the presence of missing data and results particularly suitable for climatological applications. We assume that the series $X_t$, $t = 1, \ldots, n$ admits the following decomposition ...

4. *Authors explain that they use "a neural network model of the map F and of its Jacobian J". This is a key point of the whole paper and, in particular, of sections 2.2 and 3. In section 3 we also read that the ANN has a single layer. Authors must provide further details, such has the number of neurons of the layer, the training procedure, the global minimization algorithm used and the activation functions.*

   Thank you for the comment, we have reworked the main paragraph of section 2.2 where we explain in more detail the technical aspects. Moreover, we have added the following section in the Appendix, providing further details on neural networks. This in order not to burden the main text with too much technical information, and, at the same time, provide the necessary level of detail to the interested reader:

**A    Neural Network Models for Random Dynamical Systems**

As in Section 2.2, the dynamical system has the following representation:

$$\mathbf{X}_{t+1} = F(\mathbf{X}_t) + \mathbf{E}_{t+1}, \qquad \mathbf{X}_t \in \mathbb{R}^d, \tag{1}$$

where $\mathbf{E}_t$ is an error process in $\mathbb{R}^d$. We assume that the data $\{X_t\}$ are generated by the nonlinear autoregressive model

$$X_{t+1} = f_d(X_t, X_{t-1}, \ldots, X_{t-d+1}) + \varepsilon_t, \tag{2}$$

where $X_t \in \mathbb{R}$ and $\{\varepsilon_t\}$ is a sequence of independent random variables with $E(\varepsilon_t) = 0$ and $\mathrm{Var}(\varepsilon_t) = \sigma^2$. Also, we denote with $\mathbf{J}_t$ the Jacobian of the map, evaluated at $\mathbf{X}_t$. The model of Eq. (1) can be seen as the state–space representation of the system of Eq. (2), where $\mathbf{X}_t = (X_t, X_{t-1}, \ldots, X_{t-d+1})$ and $\mathbf{E}_t = (\varepsilon_t, 0, 0, \ldots, 0)$. We derive a consistent estimator for the map $F$ and its Jacobian through a neural networks estimator of $f_d$. "Neural Networks" are a class of nonlinear models inspired by the neural architecture of the brain [Nychka et al., 1992]. These are made up of layers that in turn have connected "neurons", which send messages and share information between each other. Layers are classified into three groups: 1) input, 2) hidden, and 3) output. The input values $\mathbf{X}_t$ are received by the *input units*, which simply pass the input forward to the hidden units $u_j$. Each connection (indicated by an arrow) performs a linear transformation determined by the *connection strength* $\omega_{ij}$ so that the total input to unit $u_j$ results

$$\sum_{i=1}^{d} \omega_{ij} X_{t-i+1} + \omega_{0j} \tag{3}$$

and each unit performs a nonlinear transformation on its total input:

$$u_j = \psi \left( \sum_{i=1}^{d} \omega_{ij} X_{t-i+1} + \omega_{0j} \right). \tag{4}$$

The *activation function* $\psi$ is sigmoidal function with limiting values 0 and 1 as $x \to -\infty$ and $+\infty$, respectively. Here we adopt the following:

$$\psi(x) = \frac{x(1 + |\frac{x}{2}|)}{2 + |x| + \frac{x^2}{2}} \tag{5}$$

The hidden layer outputs $u_j$ are passed along to the *single output unit*, which performs an affine transformation on its total input. Therefore, the network output $f_d$, for $d$ inputs and $h$ units in the hidden layer, can be represented as:

$$f_d(\mathbf{X}_t) = f_d(X_t, X_{t-1}, \ldots, X_{t-d+1}) = \beta_0 + \sum_{j=1}^{h} \beta_j u_j$$

$$= \beta_0 + \sum_{j=1}^{h} \beta_j \psi \left( \sum_{i=1}^{d} \omega_{ij} X_{t-i+1} + \omega_{0j} \right).$$

We made use of the R `nnet` package to implement the estimator and we used Least Squares minimization [Venables and Ripley, 2002]. We also experimented with conditional maximum likelihood without noticing major discrepancies in the results. A practical difficulty in regression with neural network models is selecting among the many possible combinations of $d$ and $h$. Here we choose the best model that minimises the BIC defined as:

$$\mathrm{BIC} = \log(\hat{\sigma}^2) + \frac{\log(n)}{n}[1 + h(d + 2)], \tag{6}$$

where the error variance is estimated through the residual sum of squares $\mathrm{RSS} = n^{-1} \sum_{t=1}^{n} (X_t - \hat{f}_d(\mathbf{X}_{t-1}))^2$. Once the estimator $\hat{f}_d$ is obtained, a consistent estimator for the map $F$ and its Jacobian $\mathbf{J}$ can be derived by plug-in methods, as described in [Shintani and Linton, 2004].

5. *A lot of important details are in the appendix, especially in appendix A5. Authors ought to consider to move some materials of the appendix in the full text*

   We can look at moving some of A5 to the main text, but we are not sure that we will end up moving it. We already spent quite a long time thinking about what should go into the main text and appendix. We put into the main text what we thought to be essential and into the appendix information that's useful for those who want it but not essential for a general reader.

**References**

J. Fan, J. Meng, Y. Ashkenazy, S. Havlin, and J. Schellnhuber. Network analysis reveals strongly localized impacts of El Niño. *PNAS*, 114(29):7543–7548, 2017. doi: 10.1073/pnas.1701214114.

M. Friedrich, S. Smeekes, and J.-P. Urbain. Autoregressive wild bootstrap inference for nonparametric trends. *Journal of Econometrics*, 214(1):81–109, 2020. doi: 10.1016/j.jeconom.2019.05.006.

M. H. Glantz. *Introduction to El Niño*, page 375. Springer Nature, 2022. doi: 10.1007/978-3-030-86503-0.

R. S. Kovats, M. J. Bouma, S. Hajat, E. Worrall, and A. Haines. El Niño and health. *The Lancet*, 362(9394):1481–9, 2003. doi: 10.1016/S0140-6736(03)14695-8.

D. Nychka, S. Ellner, D. McCaffrey, and A. R. Gallant. Finding Chaos in Noisy Systems. *Journal of the Royal Statistical Society: Series B (Methodological)*, 54(2):399–426, 1992.

M. Shintani and O. Linton. Nonparametric neural network estimation of Lyapunov exponents and a direct test for chaos. *Journal of Econometrics*, 120(1):1–33, 2004.

W. N. Venables and B. D. Ripley. *Modern Applied Statistics with S*. Springer, New York, fourth edition, 2002. URL https://www.stats.ox.ac.uk/pub/MASS4/. ISBN 0-387-95457-0.

---

## Author Comment (AC2)

**Response to the Editor**

Rojo-Garibladi, Contreras–López; Giannerini; Salas–de–León; Vázquez–Guerra; Cartwright

August 21, 2023

Dear Editor,

many thanks for your constructive suggestions, please find below a pointwise reply to all of them, where your comments are in italic.

1. *the station that displays very regional characteristics'. The sentence is a bit cryptic. On the one hand, they both correlate with ENSO, but one is "very regional". The message should be more specific and accurate.*

   Thank you for the comment, we propose the following change.

   > We investigate whether there are correlations between temperatures on the Eastern South Pacific coast, influenced by the Humboldt Current System, and El Niño–Southern Oscillation (ENSO) events, using a set of 16 oceanic and atmospheric temperature time series from Chilean coastal stations distributed between 18° and 45° S. Spectral analysis indicates periodicities that can be related to both internal and external forcing, involving not only ENSO, but also the Pacific Decadal Oscillation, the Southern Annual Mode, the Quasi–Biennial Oscillation and the lunar nodal cycle. We carry out a nonlinear time series analysis. An asymptotic neural network test for chaos based on the largest global Lyapunov exponent indicates that the temperature dynamics along the Chilean coast is not chaotic. We calculate local Lyapunov exponents that characterize the short–term stability of the series. Using a cross entropy test, we find that just two stations in northern Chile, one oceanic, Iquique, and one atmospheric, Arica, present a significant positive correlation of local Lyapunov exponents with ENSO, with Iquique being the station that presents the greater number of regional characteristics that help it to correlate with ENSO differently from the rest. This work, having a large–scale study area and using time series from hitherto unused sources (naval records), reveals the nonlinear dynamics of climate variability in Chile.

2. *l. 25: "negative trend observed in the PDO" : is this a trend in the mean or in the variance ?*

   Reading the article in detail, we see that Falvey and Garreaud [2009] refer to the trend in the average temperatures. We will clarify this in the article.

3. *l. 50: "rigorously": use a more specific word. In what sense is this 'rigorous'.*

   Rigour here is a specific word; we mean mathematical rigour, so we can write that: "mathematically rigorous".

4. *equation p. 7: The equation only makes sense if $\delta$ is a line-matrix (meaning a vector), while it is defined as a norm p. 6. There is also confusion about the old / italic convention. The $J_{m-t}$ is a Jacobian (a matrix), and so should $T_M$, but $T_M$ appears both in boldface and in plain. While the development is classical, many readers might not get why only the largest eigenvalue $\nu_1$ (and not the other ones) survives under the limit $M \to \infty$.*

   Thank you for the comment, you are right and we have revised and made consistent the mathematical notation throughout the paper, esp. Sections 2.2 and 2.3. Of course $\delta$ is a vector and that was a typo. Also, we have clarified why only the largest eigenvalue survives asymptotically.

5. *p. 7: You mention the 'first approach', …. but never the 'second approach'*

see next comment

6. *Methods have not been fully presented before the Results section. At this point, the reader understands the definition of the short-term Lyapunov exponent in a dynamical system where the Jacobian is available, but not the algorithmic methods and choices that allow for the estimation of the short-term Lyapunov exponent from an empirical (uni-variate) time series, and how critical are these algorithmic choices. Little to nothing is said about the feed-forward neural network model (l. 220) and the BIC that are central to the results.*

Thank you for the comments, we have re-worked Sections 2.2 and 2.3 and inserted a new Section in the Appendix dedicated to neural networks modelling of random dynamical systems, see our reply to point 4. of Referee 1.

7. *p. 12: "this does not happen". Not clear what "this" refers to (negative LLE for all time series ?). Using positive wording (LLEs are positive for AST) might help.*

We have clarified as follows:

Boxplots of the LLEs versus steps ahead are shown in Fig.6 and 7. Clearly, the sea surface temperature series approach the global exponent faster than the atmospheric series. For instance, for $M = 28$ steps ahead, the boxplots of the LLEs are already negative for all the SST series, with the exception of Puerto Montt; Fig.6.On the contrary, for $M = 28$ all AST time series still have positive LLEs.

8. *The discussion on p.15 is not easy to follow. I understand that LLE may increase or decrease during an El-Niño event, and this depends on the epoch. l. 259 one refers to a "marked decrease" during the 192-1973 event but the reader will not know how well it is "marked" and what this means.*

We have added the relevant figures to facilitate the comphrehension. Please see the following revised paragraph:

In the case of the atmospheric stations, that in Antofagasta, the 1 month window shows unstable behavior throughout the entire record and shows a temperature change (higher to lower) during the 1972–73, 1982–83 and 1997–98 El Niño events (see the highest temperature peaks in Fig.D4c). In the case of Arica, for the same window, a decrease in the unstable part is observed for the same El Niño events plus that of 2014–16 (see the highest temperature peaks in Fig.D4a). For its part, Puerto Montt, in its 1 and 6 month windows, behaves in the same way as Antofagasta in its oceanic record, that is, it shows continuous and unstable behavior throughout the entire time series (see Fig.D1h and D4h). The same occurs with Rodelillo for the 1 month window (see Fig.D4e); its behavior is similar to that of Antofagasta and Puerto Montt. Finally, for the 1 year window in Serena, instability seems to increase from 1971 to 1982–83, which is where it coincides with the El Niño event of that year, the behavior decreases and increases again to coincide with El Niño of 1987–88 and likewise, a peak of instability coincides with El Niño of 2014–16 (see Fig.D6d).

9. *More broadly, I would strongly encourage the authors to reconsider their choices of figures. On the one hand, the boxplots (Fig. 6) linking the short-term to long-term Lyapunov exponents are interesting but perhaps they can go in the appendix or supplementary material. By contrast, the presentation p. 15 describing the evolution of indices through time must be supported by figures (especially given that, l. 273, you acknowledge that the result presentation so far lays upon a "visual comparison", which the reader cannot reproduce).*

Thank you, we have inserted the relevant figures, see the following revised paragraph.

In the case of the oceanic stations, we may note the following by a visual comparison: in Antofagasta the expected behavior is observed, that is, the whole series shows continuos instability for the 1 and 6 month windows, always showing the same behavior throughout the record (see Fig.D1c). This may be due to the warm water pool that is always found here simulating an El Niño. The same continuos pattern is observed for the 1 year window, only this time the behavior is completely stable (see Fig.D3c). Arica shows instability throughout the record for the 1 month

window, where instability decreases during El Niño events of 1953–54, 1958–59, 1963–64, 1965–66, 2009–10 and 2014–16 (see Fig.D1a). During the 6 month and 1 year windows, it becomes more stable than unstable and shares the same behavior with Talcahuano, where the third cut can be located lower than the other two and a very marked variability in the LLE can be seen (see Fig.D2a, f and D3a, f). In these two stations the trajectory of the phase space for the second cut is interrupted (see Fig.4). In Talcahuano, interesting behavior is only observed, with respect to the ONI, for the 1 month window, where the instability for the first cut, 1949–1974, decreases during El Niño events of 1953–54, 1958–59, 1963–64, 1965–66, 1968–69 and 1972–73, to show rising behavior in the third cut of 1991–2020 during El Niño events of 1991–92, 1994–95, 1997–98; 2002–03, 2004–05, 2009–10 and 2015–16 (see Fig.D1f). In the case of Iquique, the 1 month and 1 year windows have stable/unstable behavior, it is the only series in which the peaks of both the stable and unstable part coincide with several El Niño events of 1987–88, 1991–92, 1994–95, 1997–98, 2002–03, 2004–05, 2006–07, 2009–10, 2014–16, 2018–19, 2019–20 (see Fig.D1b and D3b). For the 6 month window, the behavior is stable throughout the record, but the stability peaks continue to coincide with the same El Niño events, with the exception of 2002–03, 2004–05, 2006–07, 2018–19 and 2019–20 (see Fig.D2b). Finally, Valparaíso, in the 1 month window, shows a decrease in the unstable part that corresponds to El Niño events of 1968–69, 1972–73, 1982–83, 1997–98 and 2014–16 (see Fig.D1e).

In the case of the atmospheric stations, that in Antofagasta, the 1 month window shows unstable behavior throughout the entire record and shows a temperature change (higher to lower) during the 1972–73, 1982–83 and 1997–98 El Niño events (see the highest temperature peaks in Fig.D4c). In the case of Arica, for the same window, a decrease in the unstable part is observed for the same El Niño events plus that of 2014–16 (see the highest temperature peaks in Fig.D4a). For its part, Puerto Montt, in its 1 and 6 month windows, behaves in the same way as Antofagasta in its oceanic record, that is, it shows continuous and unstable behavior throughout the entire time series (see Fig.D1h and D4h). The same occurs with Rodelillo for the 1 month window (see Fig.D4e); its behavior is similar to that of Antofagasta and Puerto Montt. Finally, for the 1 year window in Serena, instability seems to increase from 1971 to 1982–83, which is where it coincides with the El Niño event of that year, the behavior decreases and increases again to coincide with El Niño of 1987–88 and likewise, a peak of instability coincides with El Niño of 2014–16 (see Fig.D6d).

10. *The spectral analysis section 4.1 should be part of the result presentation, not the discussion.*

    At present this section is both results and discussion. We then should separate them, with the main findings in the results section with all the spectral peaks obtained for each station specified and here in section 4.1 adding possible explanations to the periods found and in which other studies these periodicities have appeared.

11. *Attributing spectral peaks to forcing or processes (lunar perigee subharmonic, QBO) is a delicate task, with significant risks of misattribution. Using the conditional (with the magic phrase "may be") is a wise precaution, but somewhat unsatisfactory. For example, in this particular case, can you check the phase coherence of the signal with these processes?*

    Thank you for the comment. We are aware this is a bit speculative but we support it by citing several articles about sedimentation data where high energy peaks can be related to the QBO. Our analysis is similar to that of the works we mention as we find in our data periods that have been already associated to QBO. Of course this is a correlative analysis and we will investigate the relationship between the QBO and the spectral peaks in a future work and give the discussion of this topic the space it deserves.

12. *The discussion of the nonlinear analysis (section 4.2) is also a bit tedious, some would say speculative. It is fine to formulate hypotheses, but consider the following advices: - refer to figure numbers when you describe a specific feature (to help the reader make its own judgments. For example, when you write "it is also possible to observe similar dynamics", "it can be noted by eye")*

    See 17 below.

13. *Discuss and describe possible ways to verify your hypotheses. What would it take to verify the teleconnection processes; verify the role of the atmospheric teleconnections, the role of the Pacific anticyclone, or the fact that disturbances coming from the equator only influence the northern part of Chile).*

    We will mention the matter. In general this will require a lot more data, and would be for future studies to look at.

14. *avoid non-necessary conditional (e.g.: "it is also possible to observe" : you observe it, or you don't).*

    Thanks, we will check and amend.

15. *l. 412 : "but rather Southern Annual Mode" : the sentence is unclear.*

    Yes, it should be written the other way round:

    > As this station is located south of the southern limit influenced by El Niño, we can postulate that El Niño is not responsible for the climate variability that may occur in this station, but rather Southern Annual Mode (SAM) (González-Reyes, 2013), since we can see an unstable dynamics also in the oceanic part (Puerto Montt–SST).

16. *l. 450 : a table with cross-entropies would help.*

    Thanks, we can add this.

17. *These line-by-line comments bring me to the broader editorial concern. - Overall, I consider that you need a better discussion strategy for section 4.1. The section is lengthy, and it leaves the reader unsatisfied because many hypotheses are expressed but the reader will struggle to make his/her own opinion. -Once you have presented the methods (and again, consider that at the reader lacks algorithmic details, with a discussion of the critical aspects of the algorithmic choices) my advice would be to focus on fewer elements that emerge out of your analysis, but treat them in a way that effectively helps future studies to build on them. -Remember that ESD's focus on the functioning of the Earth system. This is not the journal for presenting tons of data nor focusing on overly local aspects.*

    [This must mean 4.2 rather than 4.1.] There's a saying about working more on a text to make it shorter — i.e., it needs time and effort to condense ideas into a shorter text. We'll do our best to improve section 4.2 by shortening it.

**References**

M. Falvey and R. D. Garreaud. Regional cooling in a warming world: recent temperature trends in the southeast Pacific and along the west coast of subtropical South America (1979–2006). *Journal of Geophysical Research: Atmospheres*, 114(D04102):1–16, 2009. doi: 10.1029/2008JD010519.

---

## Author Response (AR1)

1. Regarding moving material from the appendix to the main text (suggestion of reviewer #1): The reviewer suggestion is helpful but I also agree with your position to keep the main text as focused as possible. I am fine with keepnig the appendix as it is.

R= Thanks for the positive comment to this part which remained unchanged in this new version, except for the additional material that was necessary due to the request of both reviewers for a more detailed explanation of the methods.

2. I am looking forward to read the revised section 4.

R= In this version of the article, we have added part 4 already worked on, with some additional comments to the 1.5-year cycle, of which we found some recently, curiously analyzing some results of work with paleoclimates.

3. other minor comments will be accounted for in the way you already suggested.

R= We have taken into account all the referees' comments, always focusing on improving our work with the constructive criticism that you provide us.

---

## Author Response (AR2)

1. Check l. 532 (in the 'trakcked-change file), sentence starting with "For the atmospheric stations..." : sentence is ill-formed

R = We have made the requested change